# SVEP1 is an endogenous ligand for the orphan receptor PEAR1

Jared S. Elenbaas [1,2] ✉, Upasana Pudupakkam[1], Katrina J. Ashworth [3], Chul Joo Kang[4], Ved Patel[1], Katherine Santana [1], In-Hyuk Jung[1], Paul C. Lee[1,2], Kendall H. Burks [1,2], Junedh M. Amrute [1,2], Robert P. Mecham [5], Carmen M. Halabi [5,6], Arturo Alisio [1], Jorge Di Paola [3] & Nathan O. Stitziel [1,4,7] ✉

Sushi, von Willebrand factor type A, EGF and pentraxin domain containing 1 (SVEP1) is an extracellular matrix protein that causally promotes vascular disease and associates with platelet reactivity in humans. Here, using a human genomic and proteomic approach, we identify a high affinity, disease-relevant, and potentially targetable interaction between SVEP1 and the orphan receptor Platelet and Endothelial Aggregation Receptor 1 (PEAR1). This interaction promotes PEAR1 phosphorylation and disease associated AKT/mTOR signaling in vascular cells and platelets. Mice lacking SVEP1 have reduced platelet activation, and exogenous SVEP1 induces PEAR1-dependent activation of platelets. SVEP1 and PEAR1 causally and concordantly relate to platelet phenotypes and cardiovascular disease in humans, as determined by Mendelian Randomization. Targeting this receptor-ligand interaction may be a viable therapeutic strategy to treat or prevent cardiovascular and thrombotic disease.

Sushi, von Willebrand factor type A, EGF and pentraxin domain containing 1 (SVEP1) is a poorly characterized extracellular matrix (ECM) glycoprotein[1,2] with a striking number of human disease associations. Our interest in SVEP1 began upon finding a coding variant (p.D2702G) that associated with risk of coronary artery disease (CAD) (Fig. 1a)[3]. We then sought to elucidate the disease mechanisms of the protein and evaluate the prospect of targeting it pharmacologically[4]. The role of SVEP1 in the promotion of atherosclerosis was confirmed using mouse models, but its mechanisms of action remained elusive[4]. In addition to CAD, genetic variation within the locus containing *SVEP1* is associated with hypertension[3], type 2 diabetes[3], altered outcomes in septic shock[5], and glaucoma[6–8]. Recent studies utilizing aptamer-based multiplex protein assay plasma proteomics (SomaScan)[9] have identified additional associations of SVEP1 with human traits and diseases

including pulmonary artery hypertension[10], heart failure[11], and longevity[12,13]. Combining genomics and plasma proteomics enables causal analysis of SVEP1's role in disease using Mendelian Randomization (MR). MR of plasma SVEP1 levels has revealed causal, positive associations of SVEP1 with CAD[4], hypertension[4], type 2 diabetes[4,14], and dementia[15]. Collectively, these data strongly support a deleterious role of increased SVEP1 protein levels in human aging-related disease.

In addition to its associations with chronic disease, a recent genome-wide association study (GWAS) of platelet reactivity identified an association between a missense variant within *SVEP1* (*SVEP1* p.R229G, Fig. 1a) and platelet aggregation in response to adenosine diphosphate-stimulation (ADP)[16]. The strongest genetic association with ADP-stimulated platelet aggregation in the same GWAS was an intronic variant of *PEAR1* which encodes Platelet and Endothelial Cell

[1]Division of Cardiology, Department of Medicine, Washington University School of Medicine, Saint Louis, MO 63110, USA. [2]Medical Scientist Training Program, Washington University School of Medicine, Saint Louis, MO 63110, USA. [3]Division of Pediatric Hematology Oncology, Washington University in St. Louis, St. Louis, MO 63110, USA. [4]McDonnell Genome Institute, Washington University School of Medicine, Saint Louis, MO 63108, USA. [5]Department of Cell Biology and Physiology, Washington University School of Medicine, Saint Louis, MO 63110, USA. [6]Department of Pediatrics, Washington University School of Medicine, Saint Louis, MO 63110, USA. [7]Department of Genetics, Washington University School of Medicine, Saint Louis, MO 63110, USA. ✉e-mail: jelenbaas@wustl.edu; nstitziel@wustl.edu

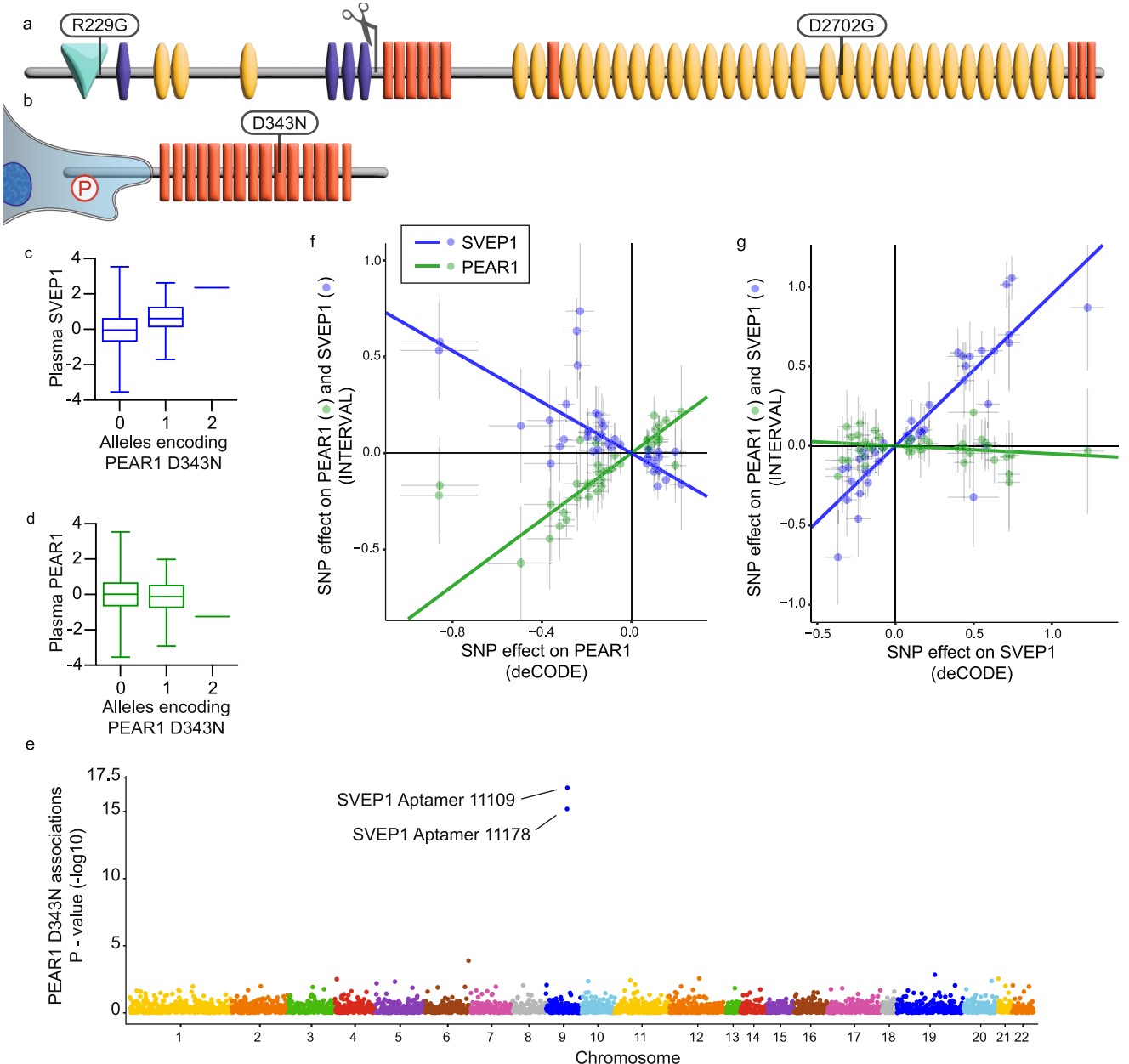

**Fig. 1 | *PEAR1* alters plasma levels of SVEP1. a, b** Schematic of SVEP1 (**a**) and PEAR1 (**b**) proteins. Domains were identified using the Simple Modular Architecture Research Tool (SMART)[84]. Teal, von Willebrand factor type A domain; purple, putative ephrin-receptor like; yellow, complement control protein/SUSHI repeat; orange, epidermal growth factor (EGF)-like domain or calcium-binding EGF-like domain or laminin-type EGF-like domain; scissors, putative cleavage site;[55] P, representative phosphorylation of the PEAR1 intracellular domain. Protein coding variants discussed in the main text are denoted at the corresponding peptide. **c** Plasma SVEP1 (aptamer 11109.56.3) as a function of allelic copies of rs147639000 (*PEAR1* p.D343N) in the INTERVAL study ($N = 3,301$). Beta = 0.67, $P = 6.5 \times 10^{-16}$. Boxes (**c, d**) depict upper and lower quartiles with median (center line); whiskers represent maximum and minimum values. (**d**) Plasma PEAR1 (aptamer 8275.31.3) as

a function of allelic copies of rs147639000 (*PEAR1* p.D343N) in the INTERVAL study ($N = 3,301$). Beta = −0.18, $P = 0.03$. **e** Manhattan plot of associations between PEAR1 D343N and 2,994 plasma proteins measured in INTERVAL. Each point represents the genomic location of the gene coding for a measured protein. **f** Two-sample MR of estimated SNP effects (with 95% confidence intervals) on PEAR1 in deCODE (*x*-axis) and either PEAR1, green, or SVEP1, blue, in INTERVAL (*y*-axis). The causal estimate is designated by a line of the corresponding color. PEAR1 Beta = 0.86, $P = 2.4 \times 10^{-37}$; SVEP1 Beta = −0.66, $P = 3.5 \times 10^{-20}$. **g** Two-sample MR of estimated SNP effects (with 95% confidence intervals) on SVEP1 in deCODE (*x*-axis) and either PEAR1, green, or SVEP1, blue, in INTERVAL (*y*-axis). The causal estimate is designated by a line of the corresponding color. PEAR1 Beta = −0.07, $P = 0.002$; SVEP1 Beta = 0.84, $P = 2.5 \times 10^{-54}$.

Receptor 1, a cell-surface receptor expressed by platelets and various vascular cells, among others[17]. Numerous additional human studies have implicated PEAR1 in platelet aggregation[18–24], as well as CAD and related outcomes[18,25–27].

PEAR1 shares many features with the receptor tyrosine kinase (RTK) family, such as dimerization, kinase activity, and likely glycosylation[17,28]; however, the PEAR1 dimer is phosphorylated by a Src family kinase (SFK)[17,29] instead of through cross-phosphorylation[30].

Antibodies that bind to the extracellular domain (ECD) of PEAR1 (PEAR1ECD) are capable of dimerizing and activating the protein, leading to its association with p85 PI3K and activation of AKT. It is speculated that PEAR1 may be a platelet binding partner[17,29] or proteoglycan receptor[31], but its function in platelets remains poorly understood. In addition, PEAR1 (also known as JEDI or MEGF12) contributes to neoangiogenesis in endothelial cells[32] and glial engulfment[33–35].

Despite the numerous disease associations of SVEP1 and PEAR1, critical gaps remain in our understanding of the molecular mechanisms of these proteins. For example, the physiological ligand of PEAR1 has yet to be identified despite previous attempts to address this critical question[28,31,36,37]. Defining the disease mechanisms of SVEP1 and PEAR1 will further our understanding of pathophysiology and may generate novel approaches to treat and prevent disease. Here, we use human multi-omics, animal models, and cellular and molecular assays to identify SVEP1 as a PEAR1 ligand, characterize their interaction, and assess the therapeutic potential of blocking these proteins.

## Results

### Plasma SVEP1 concentration is altered by *PEAR1*
To identify candidate SVEP1 interactions, we utilized previously generated human genomics and aptamer-based plasma proteomics data from the INTERVAL study of healthy volunteers[9] to conduct a genome-wide association study of plasma SVEP1 levels. Genetic variation within two loci reached a genome-wide level of statistical significance ($P \leq 5 \times 10^{-8}$; Fig. S1a). The strongest genetic association for plasma SVEP1 levels was within the locus containing *SVEP1* on chromosome 9; this cis-protein quantitative trait locus (cis-pQTL) has been described previously[4]. The only other locus that reached a genome-wide level of significance was a trans-protein quantitative trait locus (trans-pQTL) for SVEP1 on chromosome 1. The variant in the chromosome 1 trans-pQTL most strongly associated with altered plasma SVEP1 concentration (rs145662369) was an intronic single nucleotide polymorphism (SNP) within the locus containing *PEAR1*. Although rs145662369 was not associated with altered expression of *PEAR1* in the Genotype-Tissue Expression database (GTEx), in the European population it is in perfect linkage disequilibrium with rs147639000 ($r^2 = 1$, Fig. S1b), a missense polymorphism within an EGF-like domain of PEAR1's ecto-domain (Fig. 1b, p.D343N). A conditional association analysis adjusting for rs147639000 did not identify any other variants within the locus that reached genome-wide significance, suggesting that rs147639000 accounted for most of the plasma SVEP1 association within the *PEAR1* locus (Fig. S1c). Although the minor allele (Asparagine at 343) was significantly associated with increased levels of plasma SVEP1 ($P = 6.5 \times 10^{-16}$, Fig. 1c, and S1d), it was not associated with changes in *SVEP1* transcription (Fig. S1e). Plasma PEAR1 concentration was minimally altered in individuals harboring the *PEAR1* p.D343N variant ($P = 0.03$, Fig. 1d) and a proteome-wide association analysis of *PEAR1* p.D343N demonstrated that its impact on plasma protein was specific to SVEP1 among proteins measured by SomaScan (Fig. 1e). Given the impact of the *PEAR1* p.D343N variant on plasma SVEP1 concentration and because PEAR1 is expressed on vascular endothelial cells[32], we hypothesized that PEAR1 binds and sequesters circulating SVEP1 from human plasma. To test this hypothesis, we asked whether genetic variation within the *PEAR1* locus that influences plasma PEAR1 concentrations also impacts plasma SVEP1 concentrations. To avoid potential sources of confounding in a one-sample MR, we generated a genetic instrument for plasma PEAR1 and SVEP1 levels using the recently published data from deCODE, a dataset of plasma protein levels measured by SomaScan in 35,559 Icelanders[11]. Using the instruments generated from deCODE, we asked if the genetically determined plasma levels of these proteins were associated with plasma protein levels from INTERVAL. Both instruments were able to accurately predict plasma concentrations of their respective proteins (PEAR1 $P = 2.4 \times 10^{-37}$, Fig. 1f; SVEP1 $P = 2.5 \times 10^{-54}$, Fig. 1g), supporting the approach. Genetically encoded changes in plasma PEAR1 concentration were inversely related to plasma SVEP1 ($P = 3.5 \times 10^{-20}$, Fig. 1f). As expected, plasma concentrations of PEAR1 were minimally impacted by genetically encoded changes in plasma SVEP1 levels ($P = 0.002$, Fig. 1g).

We then asked whether SVEP1 and PEAR1 physically interact using molecular assays. Biolayer interferometry (BLI) was used to perform label-free, protein-binding analysis between recombinant PEAR1 and SVEP1. A global fitting of kinetic curves from an SVEP1 dilution series with biosensor-loaded PEAR1 yielded a calculated dissociation constant ($K_D$) of $8.78 \pm 0.03$ nM (Fig. 2a). SVEP1 did not exhibit appreciable binding to negative control tips, including unloaded or BSA-loaded tips (Fig. S2a). Loading of biotinylated SVEP1 to the biosensor and interferometry using a PEAR1 dilution series yielded a $K_D$ of $0.708 \pm 0.003$ nM (Fig. S2b). We then tested the hypothesis that PEAR1 D343N has reduced affinity to SVEP1 by performing interferometry with biotinylated SVEP1 and a dilution series of PEAR1 D343N. This resulted in a calculated $K_D$ of $1.76 \pm 0.006$ nM (Fig. S2c). Although this is directionally consistent with the hypothesis that PEAR1 D343N results in increased plasma SVEP1 due to decreased receptor affinity for the ligand, inherent differences between the in vitro assay and in vivo human physiology make it difficult to ascertain if the apparent 2.5-fold increase in $K_D$ is sufficient to fully explain the trans pQTL observation. The extracellular domain (ECD) of PEAR1 (PEAR1ECD) also co-immunoprecipitated with recombinant, Myc-tagged SVEP1 in pull-down assays (Fig. 2b). An alternative PEAR1ECD construct containing a biotin tag also coimmunoprecipitated with SVEP1 (Fig. S2d). Reciprocally, SVEP1 coimmunoprecipitated with the PEAR1ECD (Fig. 2c). These molecular assays and MR analyses suggest SVEP1 and PEAR1 physically interact with an affinity similar to tyrosine kinases and their ligands[38].

### *SVEP1* and *PEAR1* are co-expressed in human tissues
SVEP1 circulates in human plasma, but the protein is thought to primarily reside within the ECM of the tissues where it is produced, similar to other ECM proteins[39]. PEAR1 also acts locally as a receptor that signals intracellularly. We therefore sought to identify tissues that may harbor a biologically relevant interaction between SVEP1 and PEAR1 by determining which tissues co-express their transcripts. The expression of *SVEP1* and *PEAR1* is highly correlated among tissues in GTEx (Fig. 2d) and several tissues express high levels of both genes. For example, arterial and adipose tissues (orange and purple, respectively) express *SVEP1* and *PEAR1* and are particularly relevant to cardiometabolic disease. Bone marrow is not among the tissues analyzed in GTEx; however, other sources of expression data indicate high expression of *SVEP1* and *PEAR1* within this tissue[40]. Single-cell RNA analysis of coronary arteries[41], the site of atherosclerosis that can lead to myocardial infarction, reveals that *SVEP1* is expressed predominantly by fibroblasts (Fig. S2e), although studies in mice suggest that VSMCs may also express *Svep1* under pathological conditions[4]. *PEAR1* is expressed by a variety of disease-relevant cell-types within coronary arteries, including fibroblasts, smooth muscle cells, and endothelial cells (Fig. S2e). To assess protein expression of PEAR1, we collected platelet lysates from freshly isolated human platelets and cultured primary human umbilical vein endothelial cells (HUVECs), primary human coronary artery smooth muscle cells (hCASMCs), and 293 T cells. Immunoblot assays for PEAR1 revealed that platelets, HUVECs, and hCASMCs express the protein. In contrast, 293T cells do not express an appreciable amount of PEAR1 (Fig. 2e).

Receptors on the surface of cells interact with the ECM and influence cell behavior. The ECM is heterogenous and has disparate effects on cells. Although SVEP1 is a canonical component of the ECM, antibodies that reliably recognize SVEP1 in situ have not been developed; therefore, little is known about how and where SVEP1 may integrate into the ECM. We used a combination of affinity and proximity-based experimental approaches to address this question. The bait proteins for these experiments included recombinant SVEP1 fused to a Myc-tag or mini-Turbo ID (mTID), a promiscuous biotin ligase[42]. The prey proteins were derived from enriched media from murine VSMCs (Fig. 2f). Two independent experiments were performed using each approach and a reproducibility criterion of $P < 0.10$ for enrichment was applied across all experimental data. A total of 8 proteins fulfilled this criterion (Fig. 2g), including Basement

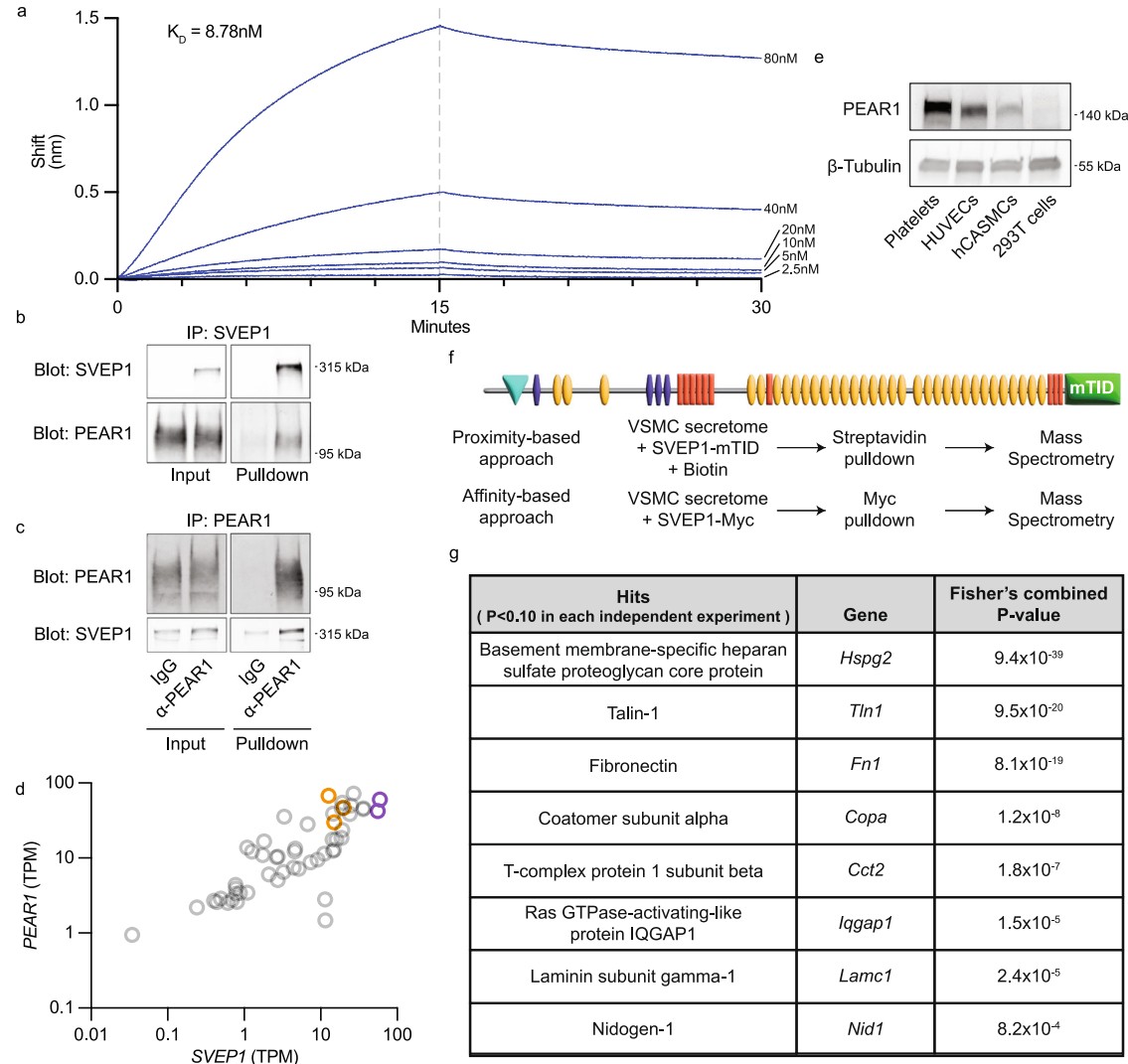

**Fig. 2 | SVEP1 and PEAR1 physically interact and colocalize in tissue. a** Biolayer interferometry sensorgrams. A dilution series of SVEP1 was analyzed by sensors loaded with PEAR1ECD. The dashed line represents the end of the association step and the beginning of the dissociation step. **b**, **c** Immunoblots of the indicated proteins after co-immunoprecipitation. Negative controls included no SVEP1 (**b**), or non-specific IgG (**c**). Additional details listed in "Methods" section. **d** Expression of *PEAR1* and *SVEP1* in transcripts per million (TPM) in tissues from the GTEx database. Purple circles designate adipose tissues. Orange circles designate arterial tissues. Pearson *r* correlation = 0.74, $P = 3.5 \times 10^{-10}$. **e** Immunoblot analysis of PEAR1 levels using cell lysates from the indicated cell-type. β-Tubulin served as a loading control. **f** Schematic of proximity or affinity-based proteomics experiments. **g** List of proteins enriched in experiments represented in **f**. Hits were identified as those proteins enriched at a confidence level of $P < 0.10$ in each experiment. Fisher's combined *p*-value is a meta-analysis of the four experiments. Additional details listed in "Methods" section.

membrane-specific heparan sulfate proteoglycan core protein (HSPG2, also known as Perlecan), Fibronectin, Laminin subunit gamma-1, and Nidogen-1. Pulldown of Fibronectin by SVEP1 was confirmed using coimmunoprecipitation assays (Fig. S2f). Together, these proteins comprise the major non-collagen basement membrane components[43]. This suggests SVEP1 may integrate with the basement membrane and potentially interact with numerous PEAR1-expressing cells.

## SVEP1 signals through PEAR1 to activate AKT signaling

PEAR1 is phosphorylated by SFK upon its activation[17,29]. To test if SVEP1 can induce PEAR1 activation and phosphorylation, we exposed platelets to immobilized bovine serum albumin (BSA, a non-specific negative control protein), immobilized SVEP1, or soluble PEAR1 polyclonal antibody (pAb, a positive control)[28,29]. Immunoblot assays revealed a robust phospho-tyrosine signal corresponding to 140 kDa, the expected mass of PEAR1, after pulling down PEAR1 from lysates of cells exposed to SVEP1 and PEAR1 pAb

but not BSA (Fig. 3a); this result is consistent with activation of PEAR1 by SVEP1[17]. We then tested activation of downstream AKT signaling by probing the platelet lysates for phosphorylated AKT (pAKT)[29]. Consistent with PEAR1 activation, both SVEP1 and PEAR1 pAb induced AKT phosphorylation in platelets, but BSA did not (Fig. 3b). We then tested the response of PEAR1-expressing HUVECs and hCASMCs (Fig. 2e) to SVEP1 using similar techniques. Serum-containing media was used in these signaling assays as a PEAR1-independent, positive control for AKT signaling. AKT signaling was activated upon exposure to SVEP1, PEAR1 pAb, and serum-containing media in both cell types (Fig. 3c, d), relative to BSA controls. Neither SVEP1 nor PEAR1 pAb activated AKT in 293T cells, which lack PEAR1 (Fig. 3e). However, serum activated AKT signaling in 293T cells, suggesting the AKT signaling axis was uncompromised in these cells. Reconstitution of PEAR1 in 293T cells by transfection of a *PEAR1*-expression plasmid resulted in constitutive AKT activation, such that SVEP1, PEAR1 pAb, and serum had no

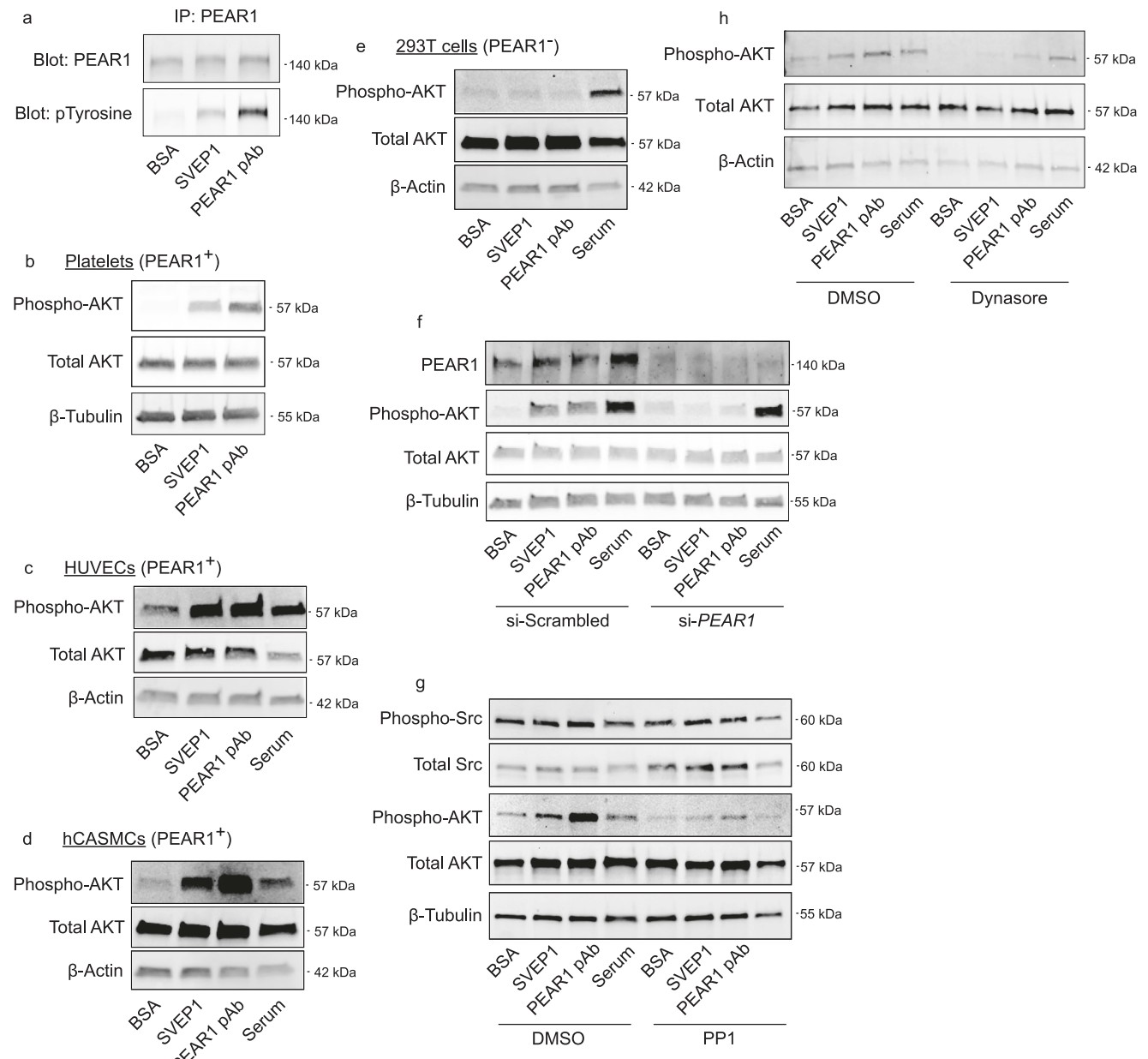

**Fig. 3 | SVEP1 activates AKT signaling through PEAR1. a** Isolated human platelets were exposed to immobilized BSA, SVEP1, or soluble PEAR1 pAb for 15 min prior to lysis. Lysates were subjected to immunoprecipitation with an anti-PEAR1 antibody and analyzed by immunoblot assays for PEAR1 and pTyrosine. The pTyrosine signal directly overlapped with the PEAR1 signal at approximately 140 kDa. **b** Platelets were exposed to stimuli as described in (**a**). Lysates were analyzed by immunoblot assays for the indicated proteins. **c**–**e** HUVECs (**c**), hCASMCs (**d**), and 293T cells (**e**) were exposed to stimuli before lysis and analysis by immunoblot assays for the indicated proteins. **f** hCASMCs were transfected with scrambled siRNA or anti-*PEAR1* siRNA prior to exposure to the listed stimuli. Lysates were analyzed by immunoblot assays for the indicated proteins. **g**, **h** HUVECs were pretreated with DMSO (carrier), PP1 (SFK inhibitor, **g**), or Dynasore (dynamin inhibitor, **h**) prior to exposure to the listed stimuli. Lysates were analyzed by immunoblot assays for the indicated proteins.

additional effect on pAKT levels in these cells (Fig. S3a). Together, these findings are consistent with the hypothesis that SVEP1 signals through PEAR1 to activate AKT. To directly test this hypothesis, we performed transient PEAR1 knockdown in hCASMCs using small-interfering ribonucleic acid (siRNA), since hCASMCs express PEAR1 and are readily transfectable with siRNA. Cells treated with *PEAR1* siRNA had diminished PEAR1 protein levels compared to negative controls (Fig. 3f and S3b) and were unable to activate AKT upon exposure to SVEP1 and PEAR1 pAb. Serum-containing media was able to activate AKT signaling regardless of siRNA treatment, demonstrating an intact AKT signaling axis. These data demonstrate that SVEP1-induced AKT signaling is dependent on PEAR1.

Activation of AKT signaling by PEAR1 is SFK-dependent[29]. To test whether SVEP1-induced AKT signaling was also dependent on SFK, we pretreated HUVECs with the SFK inhibitor PP1 or carrier dimethyl-sulfoxide (DMSO) prior to BSA, SVEP1, PEAR1 pAb, or serum exposure. As expected, PP1 abrogated the ability of SVEP1 and PEAR1 pAb to activate AKT signaling in HUVECs (Fig. 3g). PEAR1 signaling is also thought to depend on its internalization through a clathrin-dependent mechanism[33,34]. To test whether SVEP1-induced AKT signaling was also dependent on this process, we treated cells with the dynamin inhibitor Dynasore[44] or carrier DMSO prior to exposure to the stimuli. HUVECs pretreated with DMSO exhibited increased pAKT upon exposure to SVEP1, PEAR1 pAb, and serum; however, the effects of SVEP1 and PEAR1

pAb were abrogated by Dynasore pretreatment (Fig. 3h). These data demonstrate that the effects of SVEP1 on pAKT are dependent on SFK and endocytosis, consistent with canonical PEAR1 signaling.

## PEAR1 and pAKT colocalize to the lamellipodia of cells grown on SVEP1

Previous studies have reported that PEAR1 is localized to the filopodia and lamellipodia of cultured cells[32]. These cellular structures are generated by actin polymerization and are sites of membrane protrusion and ECM adhesion[45]. To test whether SVEP1 activates pAKT at regions of PEAR1 localization, we seeded hCASMCs and HUVECs on SVEP1 and stained for filamentous actin (fActin), PEAR1, and pAKT. Cells were imaged using fluorescent confocal microscopy and filopodia and lamellipodia were identified as bundles of fActin on the perimeter of the cells[45]. hCASMCs treated with scrambled siRNA exhibited high colocalization of PEAR1 and pAKT on lamellipodia and lower colocalization within the cell body, a negative control region (Fig. 4a, b). Similar colocalization was observed in HUVECs (Fig. S3c, d). Knockdown of PEAR1 using siRNA diminished the colocalization in filopodia and lamellipodia (Fig. 4a, b). Together these data suggest that PEAR1 on the surface of lamellipodia and filopodia activates pAKT locally when cells encounter immobilized SVEP1.

## SVEP1 and PEAR1 activate downstream mTOR signaling

AKT is a central regulator of numerous signaling pathways; however, little is known about which pathways downstream of AKT are activated by PEAR1. We screened for AKT-related pathways that may be influenced by SVEP1/PEAR1 signaling using an AKT pathway phospho-array. Given the temporal nature of kinase activation, we exposed HUVECs to BSA or SVEP1 for either 10 or 30 min before lysing the cells and assessing pathway activation. Elevated pAKT was observed in cells exposed to SVEP1 in each experiment (Fig. S3e), validating the methodology. Few changes were observed after 10 min of SVEP1 exposure; however, multiple phospho-proteins in the mammalian target of rapamycin (mTOR) signaling pathway were elevated after 30 min of SVEP1 exposure, including p70S6K, RPS6, and 4E-BP1 (Fig. S3e). Immunoblot assays of phosphorylated mTOR (Ser 2448) and the mTOR-regulated residue Thr 389 of p70S6K[46] further support an activation of mTOR signaling by SVEP1 (Fig. 4c). Phosphorylation of p70S6K on residue 389 was also increased by the PEAR1 pAb and serum after 30 min of exposure (Fig. 4c). Transient knockdown of PEAR1 by siRNA abrogated mTOR activation by SVEP1, as determined by immunoblot assay of p70S6K phospho-Thr 389 relative to total p70S6K in hCASMCs (Fig. 4d and S3f). Exposure of platelets to SVEP1 had similar effects on p70S6K Thr 389 (Fig. 4e), consistent with mTOR activation. Small molecule inhibitors of SFK (PP1), endocytosis (Dynasore), AKT (MK-2206), and mTOR (Rapamycin) were added to platelets prior to SVEP1 exposure to test whether SVEP1-induced AKT/mTOR signaling was dependent on the respective protein or cell process. Activation of AKT by SVEP1 was completely abrogated by inhibition of SFK and AKT and partially abrogated by endocytosis inhibition. Phosphorylation of p70S6K Thr 389 was dependent on SFK and mTOR and partially dependent on endocytosis and AKT (Fig. 4e). Taken together, these data suggest that activation of PEAR1 by SVEP1 induces AKT and downstream mTOR signaling.

## Loss of Svep1 in mice is cardiometabolically well-tolerated

AKT/mTOR signaling is activated by SVEP1/PEAR1 and plays a critical role in numerous physiologic and pathologic processes, particularly processes related to metabolism and cardiovascular function. Heterozygous Svep1 deficiency in mice appears to be well-tolerated into adulthood[4], but mice lacking Svep1 during development fail to survive past birth and exhibit marked edema in utero[47,48]. Given the biological role of AKT/mTOR signaling, the co-expression of SVEP1 and PEAR1 in adipose and vascular tissue, the disease associations of SVEP1, and the

interest in pharmacologically targeting SVEP1, we sought to characterize the chronic impact of complete SVEP1 depletion on cardiometabolic phenotypes in post-developmental mice. Six-week-old $Svep1^{flx/flx}Rosa26\text{-}Cre^{ERT2}$ (referred to as $Svep1^{-/-}$) and control littermate $Svep1^{+/+}Rosa26\text{-}Cre^{ERT2}$ (referred to as $Svep1^{+/+}$) mice were injected intraperitoneally with tamoxifen to delete SVEP1, as done previously[4]. Mice were fed a Western high-fat, high-cholesterol diet (HFD) beginning at 8 weeks of age to induce cardiometabolic stress[49–51]. Both $Svep1^{+/+}$ and $Svep1^{-/-}$ mice gained body mass while being maintained on HFD. No appreciable differences were observed in body mass between the two genotypes of mice throughout the duration of HFD feeding (Fig. S4a). Similarly, compared to littermate controls, $Svep1^{-/-}$ mice had no appreciable differences in lean mass, fat mass, and total water, as determined by EchoMRI™ (Fig. S4b–d). $Svep1^{+/+}$ and $Svep1^{-/-}$ mice also had similar responses to glucose tolerance tests (GTT, Fig. S4e) and lacked insulin sensitivity, as determined by insulin tolerance tests (ITT, Fig. S4f)[49]. The metabolic activity of the mice was also tested using indirect calorimetry. Again, no significant differences were observed between $Svep1^{+/+}$ and $Svep1^{-/-}$ mice in respiratory exchange ratio (Fig. S4g) or energy consumption (Fig. S4h). Collectively, these data suggest that whole-body deletion of Svep1 in post-developmental mice with diet-induced diabetes does not result in an overt impact on body mass, body composition, glucose handling, respiratory exchange ratio, or metabolic rate and suggest that loss of SVEP1 is metabolically well-tolerated in adult mice.

Given the association of SVEP1 with hypertension, we also tested the cardiovascular manifestations of Svep1 deletion using the same mouse cohort. Arterial catheterization was used to measure central blood pressure and heart rate in anesthetized mice. We did not appreciate significant differences between $Svep1^{+/+}$ and $Svep1^{-/-}$ mice in systolic or diastolic blood pressure or heart rate (Fig. S5a–c). We further explored vascular function by titrating the $Svep1^{+/+}$ and $Svep1^{-/-}$ mice acutely with vaso-active compounds including phenylephrine, angiotensin II, acetylcholine, and sodium nitroprusside. Blood pressure was affected by the substances in a dose-dependent manner, and no significant differences were observed between $Svep1^{+/+}$ and $Svep1^{-/-}$ mice with any drug at any dose (Fig. S5d–g). Similarly, no significant differences were observed between the two genotypes in vascular compliance of the ascending aorta or carotid artery, as determined using pressure-diameter measurements on dissected tissues (Fig. S5h, i). These vascular phenotyping data support the metabolic phenotyping data and suggest the whole-body loss of SVEP1 is cardiometabolically well tolerated in adult mice.

## SVEP1 induces platelet activation

Given the human genetic associations of SVEP1 and PEAR1 with platelet reactivity, we sought to characterize platelet phenotypes of $Svep1^{-/-}$ mice and littermate control $Svep1^{+/+}$ mice. Both genotypes of mice had similar platelet counts (Fig. 5a). Platelet surface receptor CD41 was modestly lower in platelets from $Svep1^{-/-}$ mice compared to $Svep1^{+/+}$ controls in flow cytometry assays (Fig. 5b). This difference was variable between different mouse cohorts, however. To investigate platelet function, we tested the response of platelets to agonists including ADP and protease-activated receptor-4 activating peptide (PAR4-AP). Upon stimulation, we measured platelet integrin αIIbβ3 activation (using an antibody that detects its active conformation, active CD41/61) and alpha-granule secretion (using an antibody that recognizes P-selectin, CD62). ADP or PAR4-AP-stimulated platelets from $Svep1^{-/-}$ mice had significantly lower integrin activation as compared to stimulated platelets isolated from littermate controls (Fig. 5c). Similarly, ADP-stimulation resulted in significantly lower P-selectin expression in platelets from $Svep1^{-/-}$ mice, as compared to controls (Fig. 5d).

Human platelets adhered to immobilized SVEP1 under static conditions with and without the presence of 0.1 U/mL thrombin (Fig. 5e). Recombinant SVEP1 was added to freshly isolated human

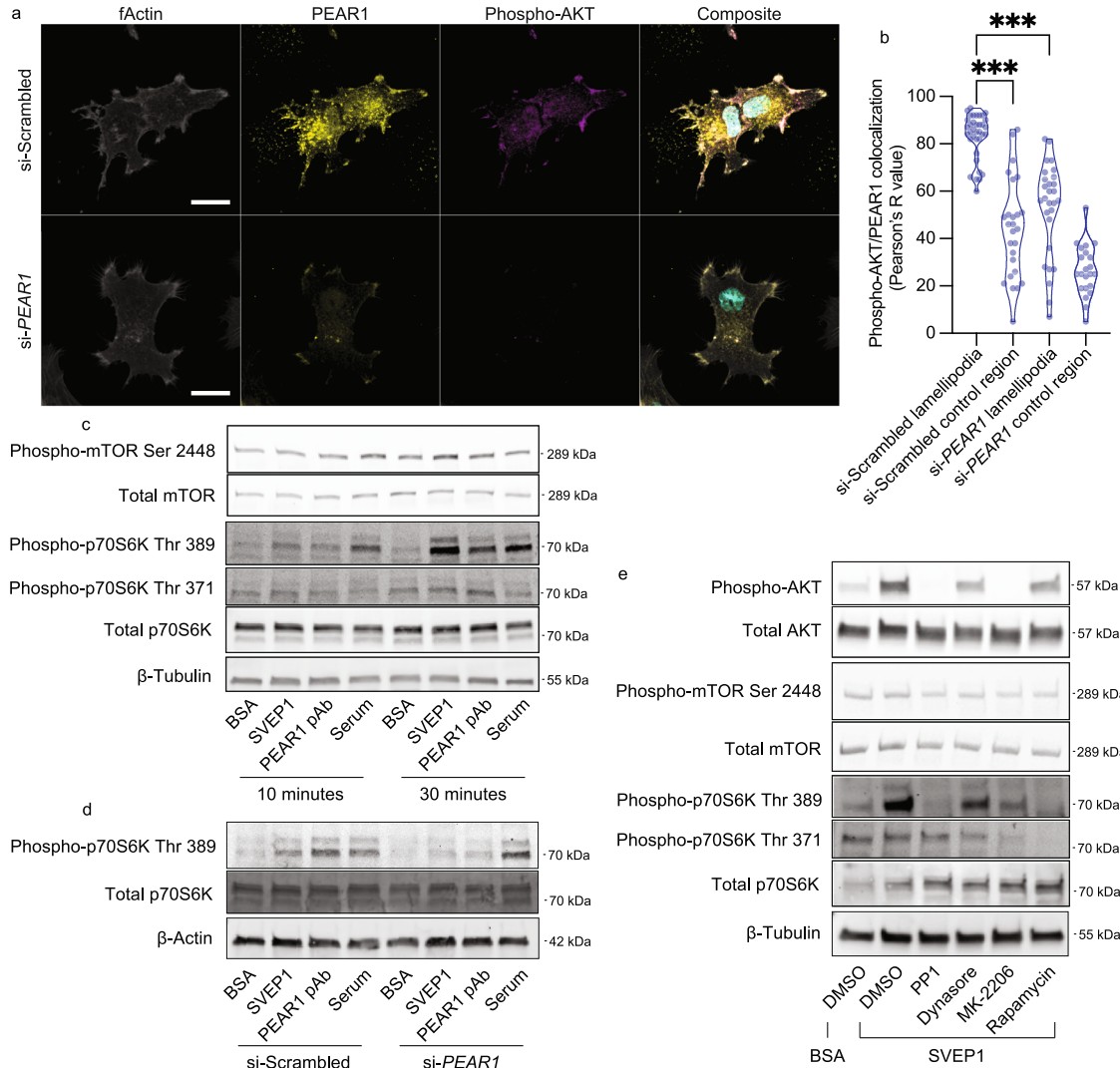

**Fig. 4 | mTOR signaling is activated by SVEP1-induced PEAR1 signaling. a** Images of hCASMCs pre-treated with scrambled siRNA or anti-*PEAR1* siRNA seeded on immobilized SVEP1 for 60 min. Scale bar = 20 µm. Composite image includes DAPI (teal). **b** Quantification of PEAR1 and pAKT colocalization in **a**, as determined by the Pearson correlation coefficient. Lamellipodia were identified as bundles of fActin on the periphery of cells. Cellular regions not containing lamellipodia were used as control regions. $N = 36, 26, 28$, and 22 regions. ***$P = 1.5 \times 10^{-15}$ or $P = 2.8 \times 10^{-11}$ from ANOVA with post hoc unpaired, two-sided *t*-test. **c** HUVECs exposed to immobilized

BSA or SVEP1, soluble PEAR1 pAb, or serum for 10 or 30 min. Lysates were analyzed by immunoblot assays for the indicated proteins. **d** hCASMCs were transfected with scrambled siRNA or anti-*PEAR1* siRNA prior to exposure to the listed stimuli. Lysates were analyzed by immunoblot assays for the indicated proteins. **e** Platelets were pretreated with DMSO (carrier), PP1 (SFK inhibitor), Dynasore (dynamin inhibitor), MK-2206 (AKT inhibitor), or Rapamycin (mTOR inhibitor) prior to exposure to BSA or SVEP1. Lysates were analyzed by immunoblot assays for the indicated proteins.

platelets or whole blood to test the effect of soluble SVEP1 on platelets. Soluble, recombinant SVEP1 induced spontaneous aggregation and agglutination of platelets in platelet rich plasma (PRP), as determined by platelet aggregometry (Fig. 5f). Platelets exposed to soluble SVEP1 had lower levels of receptor CD42b and CD61 (Fig. 5g), suggesting receptor shedding and platelet pre-activation. Platelets exposed to SVEP1 also had increased integrin αIIbβ3 activation under basal conditions and upon stimulation with ADP and Thrombin receptor-activating peptide-6 (TRAP6) (Fig. 5h). P-selectin expression was also increased in isolated platelets after addition of exogenous SVEP1 and stimulation with ADP and TRAP6 (Fig. 5i). Similar effects were observed in platelets within whole blood upon exposure to SVEP1 (Fig. S6a–c). The *SVEP1* variant (p.R229G) that associates with increased platelet reactivity in humans is also associated with increased plasma SVEP1[11,16], supporting these findings.

We then tested whether the effects of SVEP1 on platelets were dependent on PEAR1 using *Pear1*[-/-] mice[52]. The effects of SVEP1 on

platelet aggregation and agglutination were notably milder in platelets from mice (Fig. 5j) compared to humans (Fig. 5f), consistent with previous reports that murine PEAR1 plays a less prominent role in platelet function than human PEAR1[53]. Upon activation with ADP, murine platelets derived from *Pear1*[-/-] mice and incubated with SVEP1 had reduced integrin activation (Fig. 5k) and P-selectin expression (Fig. 5l) compared to platelets from control mice.

We therefore conclude, using three methodologically independent techniques (mouse models, exogenous SVEP1 assays, and human multi-omics) that SVEP1 promotes platelet activation, likely by signaling through PEAR1. Many of the effects of SVEP1/PEAR1 on platelet activation were potentiated by ADP; this finding is consistent with previous PEAR1 studies[29,53] and the GWAS associations of *SVEP1* and *PEAR1* with platelet response to ADP-stimulation in humans[16].

Given the platelet phenotypes in *Svep1*[-/-] mice, we assessed additional hematological phenotypes in these mice and found that *Svep1*[-/-] mice had higher red blood cell (RBC) counts (Fig. S6d) relative to

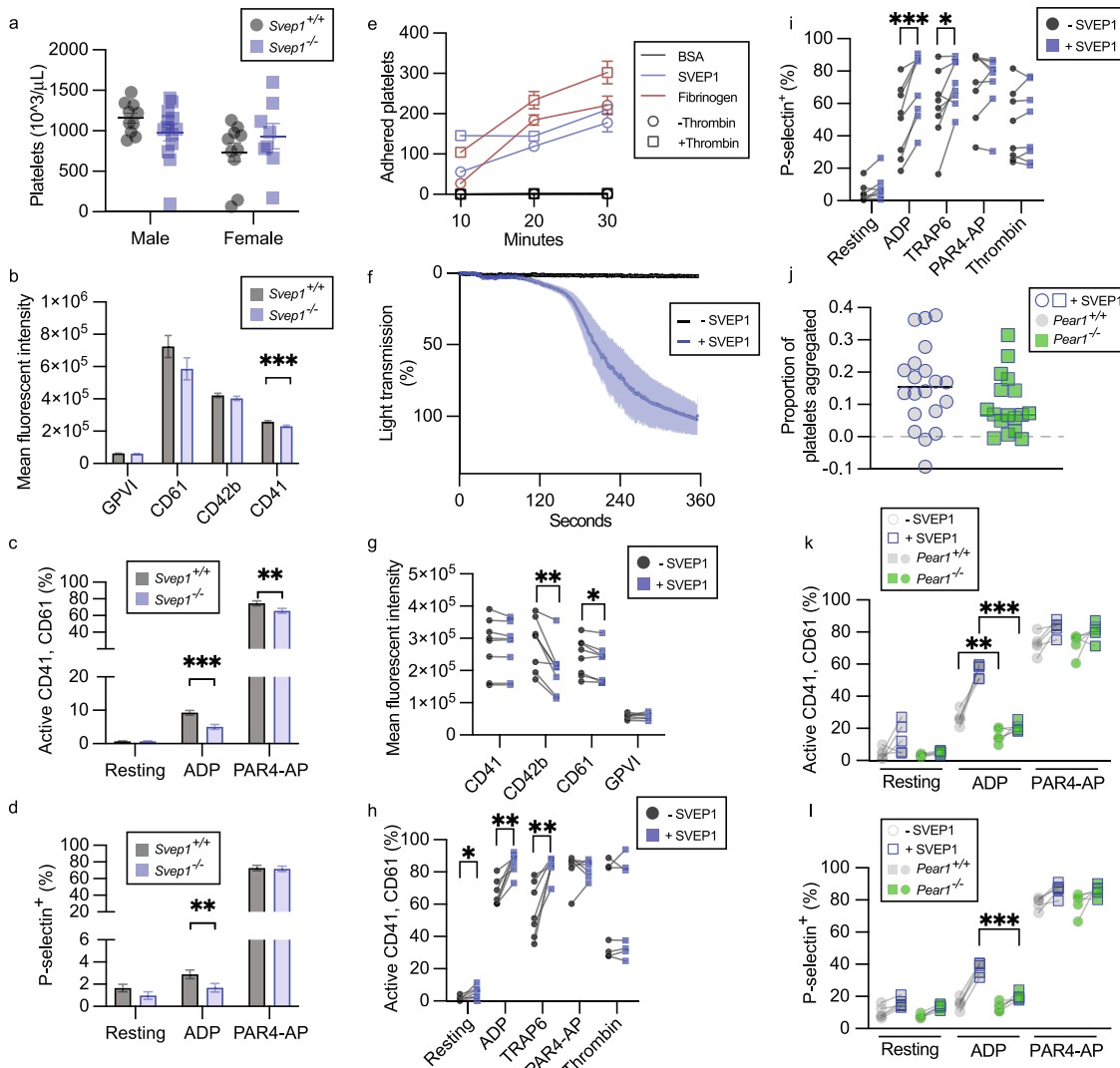

**Fig. 5 | SVEP1 activates platelets. a** Platelet counts in whole blood from *Svep1*[+/+] and *Svep1*[-/-] mice. Data are presented as mean values ± SEM. **b** Platelet receptor density in whole blood from *Svep1*[+/+] and *Svep1*[-/-] mice. Bars represent least square means and error bars represent the standard error of difference (**b**–**d**). ***$P$ = 0.0006. **c** Percentage of activated CD41/61[+] platelets from *Svep1*[+/+] and *Svep1*[-/-] mice at rest or upon exposure to the indicated stimulant. **$P$ = 0.0032, ***$P$ = 1.0 × 10[-6]. **d** Percentage of P-selectin[+] platelets from *Svep1*[+/+] and *Svep1*[-/-] mice at rest or upon exposure to the indicated stimulant. **$P$ = 0.0038. **e** Adherence of platelets to BSA (negative control), SVEP1, or Fibrinogen (positive control) coated coverslips for 10–30 min with or without thrombin. Data are presented as mean values ± SEM (**e**–**l**). **f** Aggregometry of platelet rich plasma in response to SVEP1 or carrier buffer. The shaded region corresponds to ±SEM. **g** Human platelet receptor density in freshly isolated platelets before and after exposure to SVEP1. *$P$ = 0.016, **$P$ = 0.0020. **h** Percentage of activated human CD41/61[+] platelets before and after

exposure to SVEP1 at rest or upon exposure to the indicated stimulant. *$P$ = 0.029, **$P$ = 0.0020 or $P$ = 0.0043. **i** Percentage of P-selectin[+] human platelets before and after exposure to SVEP1 at rest or upon exposure to the indicated stimulant. *$P$ = 0.027, ***$P$ = 0.0005. **j** The proportion of aggregated platelets in whole blood collected from *Pear1*[+/+] and *Pear1*[-/-] mice exposed to soluble SVEP1. **k** Percentage of activated CD41/61[+] platelets from *Pear1*[+/+] and *Pear1*[-/-] mice before and after exposure to SVEP1 at rest or upon exposure to the indicated stimulant. **$P$ = 0.0051, ***$P$ = 1.5 × 10[-6]. **l** Percentage of P-selectin[+] platelets from *Pear1*[+/+] and *Pear1*[-/-] mice before and after exposure to SVEP1 at rest or upon exposure to the indicated stimulant. ***$P$ = 8.1 × 10[-5]. *N* for groups = 10, 10, 13, 8 mice in **a**; *N* = 21, 18, 21, 17, 21, 18, 21, 18 mice in **b**; *N* = 17, 19, 20, 19, 20, 19 mice in **c**; *N* = 20, 16, 20, 17, 20, 18 mice in **d**; *N* = 3, 5, 8, 8, 8 independent human samples in **e**–**i**; *N* = 20 and 17 mice in **j**; *N* = 5 mice for all groups in **k** and **l**. Statistical significance calculated by two-way ANOVA (**a**–**d**) or two-sided *t*-test (paired for **g**–**i**, unpaired for **k**, **l**).

control *Svep1*[+/+] mice. This is consistent with genetic association of *SVEP1* with human RBC phenotypes[54]. In addition, blood from *Svep1*[-/-] mice had greater total numbers of white blood cells and lymphocytes (Figures S6e, f). Total numbers of monocytes, neutrophils, eosinophils, and basophils were not appreciably different between *Svep1*[+/+] and *Svep1*[-/-] mice (Fig. S6g–j). We then assessed whether plasma cytokine levels may explain the hematological differences between *Svep1*[+/+] and *Svep1*[-/-] mice using a cytokine array. No significant differences in plasma cytokines were observed between *Svep1*[+/+] and *Svep1*[-/-] mice (Fig. S6k), perhaps reflecting the modest effects of *Svep1* on immune cell populations.

## SVEP1 and PEAR1 are causally related to human platelet phenotypes and CAD

Finally, we asked whether SVEP1 and PEAR1 causally relate to human traits and disease. MR was used to test the impact of plasma SVEP1 and PEAR1 on mean platelet volume (MPV) and platelet count (PLT). We found that genetically determined increased plasma concentrations of both proteins associated with increased MPV (Fig. 6a; SVEP1 $P$ = 5.1 × 10[-6]; PEAR1 $P$ = 1.8 × 10[-8]) and decreased PLT (Fig. 6b; SVEP1 $P$ = 0.015; PEAR1 $P$ = 2.3 × 10[-5]). Similarly, genetically encoded changes in plasma concentrations of both proteins were positively associated with risk of cardiovascular disease (Fig. 6c; SVEP1 $P$ = 4.5 × 10[-12]; PEAR1

$P = 0.0051$). These data demonstrate that both SVEP1 and PEAR1 causally relate to platelet traits and CAD. The effects of the proteins are concordant, consistent with the hypothesis that the two proteins interact to influence disease.

## Discussion

Recent genomic and proteomic studies have implicated SVEP1 and PEAR1 in a variety of overlapping human traits and diseases. Our understanding of the mechanisms of these proteins has been limited, however, since little was known about their molecular interactions. Previous studies have reported an interaction between PEAR1 and High affinity immunoglobulin epsilon receptor subunit alpha (FcεRIα)[36]. The differing expression pattern of these two proteins, the inability of monomeric FcεRIα to activate platelets, and the lack of conservation in mouse suggests FcεRIα is not the primary physiological ligand of PEAR1[31], although pentameric FcεRIα may have a role in PEAR1 signaling[36]. Similarly, the only protein known to interact with SVEP1 is integrin α9β1[55]. Both SVEP1 and integrin α9β1 play a role in lymphangiogenesis, but $Svep1^{-/-}$ mice die much earlier than $Itga9^{-/-}$ mice (at birth vs postnatal day 14, respectively)[56]. Svep1 plays a similar developmental role in zebrafish, but $itga9^{-/-}$ larvae do not phenocopy $svep1^{-/-}$ larvae. In fact, zebrafish Svep1 lacks the putative integrin α9β1 binding domain altogether[47]. Mice lacking $Itga9$ in monocytes and smooth muscle cells also fail to phenocopy the atherosclerosis phenotype of mice lacking $Svep1$[57]. These findings suggest that SVEP1 is likely to have additional interactions related to its role in development and disease.

Here, we provide evidence that SVEP1 is a physiological ligand of PEAR1. In addition, their binding occurred with a stronger affinity than the other interactions reported for each protein[36,55]. The observation that $PEAR1$ p.D343N associated with altered plasma SVEP1 levels in humans led us to test the causal relationship between plasma PEAR1 and plasma SVEP1, since an inverse correlation would suggest PEAR1 can sequester plasma SVEP1. Indeed, genetically encoded plasma PEAR1 levels were strongly inversely correlated with plasma SVEP1 levels. SVEP1 and PEAR1 physically interacted and immobilized SVEP1 activated canonical PEAR1 signaling in a PEAR1-dependent fashion. We also found that mTOR signaling was activated downstream of SVEP1/PEAR1-induced AKT activation; these findings are summarized in Fig. 6d. It is unclear whether activation of AKT/mTOR by SVEP1/PEAR1 is directly responsible for their causal disease and trait associations; however, these pathways are well known to contribute to platelet biology[58,59], cardiometabolic disease[60-63], and longevity[64].

Several independent studies have reported associations between SVEP1 and PEAR1 in cardiovascular disease and platelet phenotypes[3,16,18,25-27], yet causality is more difficult to assess. Here we provide evidence that both proteins causally relate to human cardiovascular disease and platelet phenotypes using MR and mouse models. Mendelian Randomization can be used to test causal relationships in human biology and disease without the resource constraints and ethical limitations of clinical trials. This method relies on SNPs within a population that influence a quantitative exposure, such as plasma protein concentration, and an outcome of interest. A critical assumption of this technique is that the SNPs exclusively influence the exposure[65]. Most SNPs comprising the genetic instruments in this manuscript are non-coding; therefore, their associated differences in plasma protein concentration are likely a manifestation of the quantitative differences in protein production rather than functional differences. Proteins are known to leak from tissue to plasma and rigorous techniques have demonstrated that SVEP1 behaves in this manner[66]. Taken together, this suggests plasma protein concentration may be a proxy for tissue levels of the protein. The mechanisms of SVEP1 and PEAR1 ingress and stability in the plasma are unclear; however, the variables that regulate these processes are randomly distributed across the cohort according to Mendel's law of

independent assortment and therefore should not be a source of confounding. Given that plasma protein concentration may reflect tissue protein concentration, we conclude that the causal relationships of SVEP1 and PEAR1 described in this study are not limited to explanations pertaining to the plasma. Nevertheless, the effects of the proteins on CAD, platelet volume, and platelet count are concordant, consistent with the disease mechanisms of SVEP1 and PEAR1 being inter-related.

Several studies have independently concluded that increased SVEP1 in deleterious in humans[4,9,11,14,15]. A single study in mice contrasts these conclusions by reporting that $Svep1$ haploinsufficiency increased atherosclerosis[67]; however, the results were difficult to interpret due to confounding introduced by differing proportions of males and females in their control and experimental groups[68,69]. Our previous study avoided this source of bias and directly contradicted their conclusions using the same model in addition to complementary mouse models and outcomes[4]. SVEP1 is critical for proper development in mice[47], but our findings suggest that it may be dispensable in the adult animal, since we did not appreciate any biologically significant adverse cardiometabolic phenotypes in aged, metabolically challenged $Svep1^{-/-}$ mice. The human population variance of genetically encoded SVEP1 and PEAR1 levels suggests a safe therapeutic window exists to target SVEP1 and/or PEAR1 and potentially reduce their associated disease burden. The interaction between SVEP1 and PEAR1 occurs within the extracellular space, making this interaction an intriguing target for pharmacological intervention. Additional studies will be necessary to further characterize the mechanisms by which SVEP1 and PEAR1 influence disease and evaluate the potential of therapeutically disrupting their interaction.

## Methods
### Study approval
Blood collection from consenting healthy controls was conducted in accordance with the Institutional Review Board of Washington University, St Louis. All animal studies were performed according to procedures and protocols approved by the Animal Studies and Institutional Animal Care and Use Committees of the Washington University School of Medicine.

### Statistics and reproducibility
The specific statistical methods used to analyze each set of data are described in the figure legends and/or the specific methods section. Each measurement represents a distinct sample. One-way analysis of variance (ANOVA) was used when making multiple comparisons to a single reference group, followed by the indicated statistical test. Individual data points were shown whenever possible; however, least squared means were used to simplify data visualization in limited cases. The paired data were analyzed by a two-tailed, paired $t$-test. The unpaired data were analyzed by a two-tailed, unpaired $t$-test or a two-way ANOVA if another variable, such as sex, was a potential source of variation. Normality was assumed, and unless otherwise stated, error bars represent the standard error of mean (SEM). Data were excluded prior to analysis whenever a technical error was noted during data collection. The ROUT method under the most stringent threshold ($Q = 0.1\%$) was used to exclude outliers from the mouse hematological studies. The cell culture and molecular experiments in this manuscript were repeated independently with similar results at least one time. The BLI assay was performed three independent times with similar results. The animal experiments were performed at least once. The protein array experiments served as a screening tool and were performed once. Excluding the indirect calorimetry measurements, the animal experiments were performed in blinded and randomized fashion. The cellular studies, molecular studies, and the data analysis were performed in unblinded fashion. Densitometry of immunoblots was performed using Image Lab and reported whenever the results were not

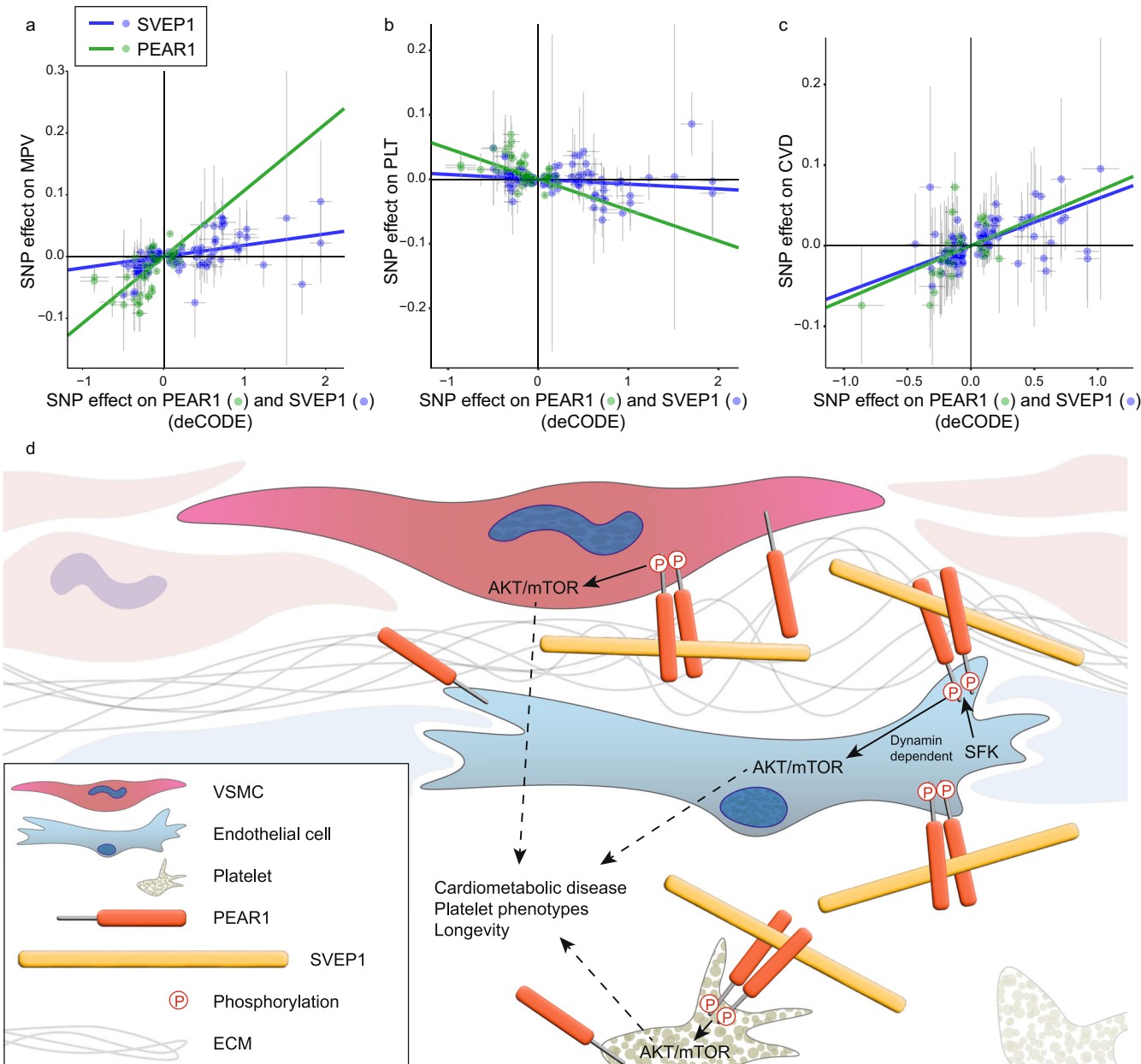

**Fig. 6 | SVEP1 and PEAR1 causally and concordantly relate to human platelet traits and cardiovascular disease. a** Two-sample MR of estimated SNP effects (with 95% confidence intervals) on MPV (y-axis) and either PEAR1, green, or SVEP1, blue, in deCODE (x-axis). The causal estimate is designated by a line of the corresponding color. SVEP1 Beta = 0.018, $P = 5.1 \times 10^{-6}$; PEAR1 Beta = 0.11, $P = 1.8 \times 10^{-8}$. **b** Two-sample MR of estimated SNP effects (with 95% confidence intervals) on platelet count (y-axis) and either PEAR1, green, or SVEP1, blue, in deCODE (x-axis). The causal estimate is designated by a line of the corresponding color. SVEP1

Beta = −0.0075, $P = 0.015$; PEAR1 Beta = −0.048, $P = 2.3 \times 10^{-5}$. **c** Two-sample MR of estimated SNP effects (with 95% confidence intervals) on cardiovascular disease (y-axis) and either PEAR1, green, or SVEP1, blue, in deCODE (x-axis). The causal estimate is designated by a line of the corresponding color. SVEP1 Beta = 0.058, $P = 4.5 \times 10^{-12}$; PEAR1 Beta = 0.067, $P = 0.0051$. **d** Model of the vascular wall and lumen. Solid lines represent experimentally tested relationships. Dashed lines represent relationships supported by indirect evidence.

abundantly clear. The data were analyzed and graphed in GraphPad PRISM v9.4.1 or R v4.0.3. The data panels were imported and formatted into figures using Adobe Illustator. Stars were used to denote statistical significance in the functional studies. *$P < 0.05$, **$P < 0.01$, ***$P < 0.001$.

### Genetic association study of plasma SVEP1 levels and Mendelian randomization
To perform a genome-wide association study of plasma SVEP1 levels, we obtained individual level genotypes and rank-inverse normalized plasma levels of 2,994 proteins as measured by SomaScan in 3,301

participants from the INTERVAL pQTL GWAS[9] (European Genome-phenome Archive Study ID EGAS00001002555). Linear regression using an additive genetic model was used to test genetic association for plasma SVEP1 with and without adjusting for rs147639000. To perform Mendelian randomization, we obtained genome-wide summary statistics for plasma levels of SVEP1 (aptamer SVEP1.11109.56.3 and aptamer SVEP1.11178.21.3) and PEAR1 (aptamer PEAR1.8275.31 chosen for its ability to detect *PEAR1* cis-pQTLs[9]) from INTERVAL[9] in addition to the previously published deCODE[11] pQTL study of 35,559 Icelandic individuals. We used unlinked GWAS markers ($r^2 \le 0.2$) from deCODE (as estimated from 1000G European sequence data[70]) to

generate instrumental variables for plasma levels of SVEP1 and PEAR1. As trans-pQTLs may affect protein levels in a variety of manners, we focused our analysis on cis-pQTLs by only including variants in a 250 kb window surrounding the gene of interest which associated with altered plasma levels of the associated protein at a level exceeding genome-wide significance (P-value for respective plasma protein concentration $\leq 5 \times 10^{-8}$). This process resulted in instruments for SVEP1 and PEAR1 which contained 43 and 40 markers, respectively, and explained 7.4% and 2.4% of the variance in their respective protein levels from INTERVAL as estimated by a genomic-relatedness-based restricted maximum-likelihood (GREML) approach.

The reported outcomes included plasma levels of proteins from INTERVAL, platelet traits, and CAD. For blood platelet traits, we obtained previously published summary statistics for platelet count and mean platelet volume in Europeans[71]. Summary statistics for CAD were obtained from the previously published meta-analysis of UK Biobank and CARIDoGRAM-PlusC4D[72]. Additional details regarding the clinical characteristics of participants in these studies and sample sizes can be found in the cited references. Causal analysis was performed using the inverse-variant weighted method implemented in the R package TwoSampleMR[73].

### Biolayer interferometry

BLI assays were performed on the ForteBio Octet RED96e and analyzed using Octet Data Analysis HT 12.02.2.29 software. Biotinylated PEAR1ECD or SVEP1, described subsequently, was loaded on Sartorius Octet SA Biosensors. Loading of PEAR1ECD was performed in the presence of 1% BSA and was stopped after all sensors reached a 1.5 nm shift. Loading of biotinylated SVEP1 was performed in a similar fashion and was stopped after 20 min of loading (corresponding to an average shift of approximately 0.7 nm). Negative controls included unloaded tips and tips loaded with biotinylated BSA. The sensors were then quenched with 5 mg/mL biocytin. After establishing a baseline in assay buffer (0.01% Tween 20 in calcium-containing DPBS), the sensors were placed in solutions of SVEP1 ranging from 2.5–80 nM for PEAR1ECD-loaded tips, 5-180 nM SVEP1 for BSA-loaded tips, or 10–90 nM PEAR1 or PEAR1 D343N for SVEP1-loaded tips. A solution containing assay buffer was used as a reference. After 15 min of association, the sensors were returned to assay buffer for 15 min of dissociation. Sensors lacking the ligands did not respond to the analytes (sensor references). The original sensorgrams with SVEP1 as the analyte were negative and flipped for analysis. The buffer reference was subtracted from the sensorgrams prior to quantification. Savitzky-Golay filtering was applied to the data. The following inter-step corrections were applied: association step (0–3 s after the start of the association), dissociation step (0–5 s after the start of dissociation). The sensorgrams were fit to a global, 1:1 binding curve, $R^2 = 0.997$-$0.999$. Local fittings provided similar dissociation constants to the global fitting.

### Immunoprecipitation

For the PEAR1 immunoprecipitation (IP): 150 μg of cell lysates from human platelets exposed to immobilized BSA, SVEP1, or a soluble polyclonal PEAR1 antibody for 15 min were incubated with 2 μg PEAR1 monoclonal antibody (R&D Systems) for 2 hours. Subsequently, 20 μL of BSA-blocked Invitrogen Protein A Dynabeads slurry was added, followed by rotation for 45 min. The beads were separated using magnetism and washed with RIPA buffer 4–5 times, then resuspended with reducing LDS sample buffer and analyzed by immunoblot assay. Bands for p-tyrosine and PEAR1 were detected at approximately 140 kDa.

For the SVEP1/PEAR1 Co-IP: Recombinant SVEP1-Myc and PEAR1ECD-Bio-His or PEAR1ECD (R&D Systems) was added to assay buffer in microcentrifuge tubes in a 3:1 mass ratio, resulting in an approximately equal molar ratio. The assay buffer consisted of 1 mg/mL BSA, 0.01% Tween 20, and 10 mM $Ca^{2+}$ in PBS. The proteins were incubated together for two hours while rotating. Where indicated, 2 μg primary antibodies were added during the final 30 min of the initial incubation. Following the incubation, aliquots were reserved as the input fraction. Subsequently, 6-10 μL slurry of Pierce Anti-c-Myc Magnetic Beads, Invitrogen Dynabeads Protein G beads, or Pierce Streptavidin Magnetic Bead slurry were added, followed by a 1-hour incubation at 20 °C with gentle agitation. Beads were washed in PBS + 0.01% Tween 20 and resuspended reducing LDS sample buffer and analyzed by immunoblot assay. Bands for SVEP1 were detected at approximately 300 kDa. Bands for PEAR1ECD were detected at approximately 105 kDa.

### Cell signaling and immunoblot assays

A list of cellular reagents is provided in Table S1. Primary human umbilical vein endothelial cells (HUVECs) were obtained from Cell Applications Inc. and cultured according to the manufacture's recommendations. Primary human coronary artery smooth muscle cells were obtained from Invitrogen and were cultured according to the manufacture's recommendations. 293T cells were obtained from ATCC and were cultured according to the manufacturer's recommendations. Predesigned Silencer Select siRNA constructs targeting PEAR1 and negative control siRNA were obtained from ThermoFisher. Transfections were performed using RNAiMAX or Lipofectamine 3000 transfection reagents according to the manufacturer's protocols. Cells were used for signaling assays 48 h after transfection; efficient PEAR1 knockdown or PEAR1 overexpression was confirmed by immunoblot assays.

Prior to performing signaling assays, cells were trypsinized, centrifuged, suspended in basal media, and counted using an automated hemocytometer. Cells were further diluted in basal media to an assay-dependent concentration. Cells were then incubated with gentle agitation for 60 min to prevent cell attachment and reduce basal signaling. PP1, Dynasore, MK-2206, and Rapamycin were diluted in DMSO and used in assay concentrations of 10 μM, 100 μM, 10 μM, and 1 μM, respectively, for the final 20-30 min of incubation in basal media. An equal volume of DMSO was used as the negative control condition. 1 mL of the cell-culture was then seeded on 24-well tissue culture plates pre-coated with 15–30 μg/mL BSA or SVEP1 and washed with Dulbecco's phosphate-buffered saline (DPBS). 1 μg PEAR1 polyclonal antibody was added as a specific positive control. In all, 20% growth media or 2% fetal bovine serum, labeled "Serum" in figures, was added as a non-specific positive control. The cells were centrifuged at 300×g for 3 min with the exception of platelets, which were centrifuged at 500×g for 5 min. The cells were then incubated at 37 °C for 8–45 min, depending on the cell type and pathway of interest. The cells were lysed in radioimmunoprecipitation buffer (RIPA) containing a cocktail of protease and phosphatase inhibitors and universal nuclease. Immunoblots were performed by standard techniques, as briefly follows. Protein content was determined using a bicinchoninic acid assay with BSA standards (#23225, Pierce BCA Protein Assay Kit). Cell lysates were then reduced with dithiothreitol (DTT) in lithium dodecyl sulfate (LDS) sample buffer (#NP0007, Invitrogen). Equal protein amounts were added to polyacrylamide gels (#4561086, BioRad) and electrophoresed prior to transferring to a nitrocellulose or polyvinylidene fluoride membrane (#1620260, BioRad). Membranes were blocked in 5% BSA/Tris-buffered saline (TBS) with 0.1% Tween 20 for 30 min. The indicated primary antibodies were incubated with the pre-blocked membranes overnight at 4 °C. Membranes were washed with TBS with 0.1% Tween 20, probed with fluorescent secondary antibodies, and imaged. β-actin or β-tubulin served as a loading control.

For the protein array assays, HUVEC cell lysates were used with the C-Series Human and Mouse AKT Pathway Phosphorylation Array C1 (Raybiotech Inc.) according to the manufacturer's protocol. The signal intensity was normalized to the lysates from negative control BSA-coated wells. Plasma was pooled from two Svep1[+/+] or Svep1[-/-] mice to

constitute a single biological replicate for the mouse cytokine array C3 assay; two samples were derived from each sex of each mouse genotype for the assay.

## Cell imaging and colocalization analysis

HUVECs or siRNA transfected hCASMCs were trypsinized and seeded on chamber slides precoated with 30 μg/mL SVEP1. Cells were incubated for one hour at 37 °C, rinsed with DPBS, and fixed with 4% paraformaldehyde. Cells were washed, then blocked and permeabilized with 0.3% Triton X-100 and 5% chicken serum in TBS. Cells were incubated with PEAR1 pAb and anti-pAKT antibody for two hours, washed with TBS + 0.1% Tween 20, and incubated with secondary antibodies and phalloidin stain (for fActin), for 1 h. Chambers not treated with the primary antibody were used as a negative control. Cells were washed and treated with Prolong Diamond Antifade with 4′,6-diamidino-2-phenylindole (DAPI) overnight at room temperature. The cells were imaged the following day using confocal microscopy. The fluorescent channel for fActin was used to identify cells and focus the microscope for imaging. Composite images were split into composite pseudocolors using Fiji. The fActin channel was used to identify lamellipodia (bundles of fActin on the periphery of cells) and control, non-lamellipodia cellular regions (see Fig. S3c for representative images). The Fiji plugin Coloc 2 was used to measure intensity-independent colocalization between the PEAR1 and pAKT channels. Pearson's correlation coefficient was reported as a measure of colocalization between PEAR1 and pAKT.

## Human blood collection and platelet isolation

Whole blood was collected by venipuncture into either heparin vacutainers (BD, Franklin Lakes) for whole blood experiments or acid-citrate-dextrose (ACD) vacutainers (BD, Franklin Lakes) for platelet isolation studies. For platelet isolation, samples were supplemented with apyrase (Sigma, St Louis) and prostaglandin E1 (Cayman Chemical, Ann Arbor) and PRP was prepared by centrifugation of the ACD whole blood for 20 min at 200×g. Platelets were isolated from the PRP by centrifugation for 10 min at 1000×g and re-suspended in modified Tyrode's buffer twice at desired platelet counts and kept at 37 °C until used. Isolated platelets were used within 2 h of preparation.

## Static adhesion assays

Coverslips were pre-coated with either 15 μg/mL recombinant SVEP1, 100 μg/mL fibrinogen (as a positive control) or 1% BSA (as a negative control) in phosphate-buffered saline (PBS). Subsequently, coverslips were blocked with 1% BSA and washed with PBS. Human platelets were isolated from whole blood and re-suspended in Tyrode's buffer. In total, $2 \times 10^7$/mL basal or thrombin activated platelets were added to coverslips and incubated at 37 °C for 10, 20, or 30 min. After incubation, non-adherent platelets were removed and the coverslips were washed with PBS, fixed with 1.5% paraformaldehyde, permeabilized with 0.01% Triton-X and stained with TRITC phalloidin. Platelets were visualized using fluorescent microscopy. Images of the adhered platelets were captured using fluorescent microscopy and counted manually by a blinded observer.

## Platelet aggregometry

Human platelet rich plasma (PRP) was obtained by centrifugation of citrated whole blood at 300×g for 20 min. Platelet aggregation in response to SVEP1 (15 μg/mL) or an equal volume of carrier buffer was assayed on a PAP-8E platelet aggregometer (BIO/DATA, Horsham). ADP (10 μM) or Thrombin (0.1 U/ml) were used as positive controls (not shown).

## Blood cell counts and flow cytometry

Complete blood counts (CBC) and washed platelet counts were determined using a hematology analyzer Element HT5 (Heska,

Loveland). For flow cytometry, diluted whole blood or isolated platelets were pre-incubated with their respective fluorescent antibodies for 15 min and fixed. For murine whole blood: CD41-VioBlue, CD61-PE (Mitlenyi Biotec, Bergisch Gladbach, Germany), CD42b-DL649, GPVI-FITC (Emfret, Eibelstadt, Germany) and for human whole blood and isolated platelets: CD41-FITC, CD61-APC, CD42b-PE (Biolegend, San Diego), GPVI-BV421 (BD, Franklin Lakes). Platelet surface receptor levels were quantified by flow cytometry on a CytoFlex analyzer.

## Quantification of platelet integrin αIIbβ3 activation and P-selectin expression

Diluted human whole blood or isolated human platelets were treated with SVEP1 as described previously, pre-incubated with FITC-PAC-1 and P-selectin-PE antibodies (BD, Franklin Lakes), and stimulated with either ADP (10 μM) (Chronolog, Harverton); Thrombin receptor-activating peptide-6 (TRAP-6; 10 μM) (Tocris, Bristol, UK); Protease-activated receptor-4 activating peptide (PAR4-AP; 100 μM) (Abcam, Cambridge, UK) or Thrombin (0.1 U/mL) (Chronolog, Harverton) for 15 min. Samples were immediately fixed and run using a CytoFlex analyzer and (Beckman Coulter, Pasadena) and analysis was performed with Kaluza software (Beckman Coulter, Pasadena).

For murine whole blood flow cytometry, whole blood was collected from the retro-orbital plexus using heparinized capillary tubes. Diluted whole blood was pre-incubated with fluorescently conjugated JON/A-PE and CD62P-FITC antibodies (Emfret, Eibelstadt, Germany) and stimulated with either ADP (10 μM), PAR4-AP (100 μM) or Thrombin (0.1 U/mL) for 15 min. After incubation, samples were immediately fixed and read on a CytoFlex analyzer.

## Mice

The generation and validation of an inducible $Svep1^{-/-}$ allele and mouse model was described previously[4]. In brief, mice were generated by KOMP (Knockout Mouse Project) and crossed with mice expressing the flippase FLP recombinase under the control of the promoter of the human actin beta gene to generate $Svep1^{flx/flx}$ mice. We crossed these mice with $Rosa26$-$CreER^{T2}$ (no. 008463, the Jackson Laboratory) mice to generate $Svep1^{flx/+}Rosa26$-$CreER^{T2}$ mice. Male and female $Svep1^{flx/+}Rosa26$-$CreER^{T2}$ were crossed to generate experimental $Svep1^{flx/flx}Rosa26$-$CreER^{T2}$ ($Svep1^{-/-}$) and $Svep1^{+/+}Rosa26$-$CreER^{T2}$ ($Svep1^{+/+}$) littermate control mice. To activate Cre-recombinase, mice were injected intraperitoneally with 2.5 mg of tamoxifen (no. T5648, Sigma-Aldrich) in 0.1 mL of peanut oil (no. P2144, Sigma-Aldrich) for 5 consecutive days starting at 6 weeks of age. Tamoxifen treatment was performed with all experimental and control mice in an identical manner. Given the cardiometabolic and age-related disease associations of $SVEP1$ in humans, we used aged $Svep1^{-/-}$ and control $Svep1^{+/+}$ mice fed a western diet comprised of 21% fat by weight (42% kcal from fat) and 0.2% cholesterol (#TD.88137, Envigo Teklad) beginning at 8 weeks of age. This diet was referred to as "HFD" throughout the text. The metabolic phenotyping of these mice occurred between 8 and 9 months of age, the hematological phenotyping occurred between 10 and 12 months of age, and the vascular phenotyping occurred between 12 and 13 months of age.

The mice referred in this text as "$Pear1^{-/-}$" are the $Pear1^{tm1a(KOMP)Wtsi}$ mice generated by KOMP (generously provided by Dr. Bruce Carter, Vanderbilt University). The "$Pear1^{+/+}$" control mice are age and background matched C57BL/6NCrl mice (Charles River Laboratories) and were acclimated in the same facility as the $Pear1^{-/-}$ mice for at least one week prior to the experiments. The $Pear1^{-/-}$ and $Pear1^{+/+}$ mice were fed a standard chow diet and were assessed at 6 weeks of age. All animal studies were performed according to procedures and protocols approved by the Animal Studies and Institutional Animal Care and Use Committees of the Washington University School of Medicine. All mice were housed in the Washington University School of Medicine animal facility and maintained on a 12-h light/12-h dark

cycle with a room temperature of $22 \pm 1\,°C$ and relative humidity between 30 and 70%.

## Arterial blood pressure measurement

Central arterial blood pressure and heart rate were measured under inhaled 1.5% isoflurane anesthesia and while mice were maintained at $37\,°C$ using a heating pad and rectal thermometer, as done previously[74]. Briefly, a midline incision was performed in the neck region; the thymus, muscle, and connective tissue were dissected away to isolate the right common carotid artery. After tying it distally and clamping it proximally, an incision was made in the right common carotid artery through which a Millar pressure transducer (model SPR-1000, Houston, TX) was introduced, the clamp was removed, and the transducer advanced to the ascending aorta. Once instrumentation was complete, arterial blood pressure (systolic, diastolic, and mean) and heart rate were recorded via the PowerLab® data acquisition system (ADInstruments, Colorado Springs, CO). The average of a 3-min period of stable recording was reported. Data were analyzed using LabChart® 8 for MAC software (ADInstruments).

To assess the blood pressure response to vasoactive agents, after baseline blood pressure measurement, dissection was performed to visualize the left internal jugular (IJ) vein as done previously[75]. Once identified, a small incision was made and PE-10 tubing was introduced and kept in place with a 6-0 silk suture. While measuring arterial blood pressure, 50 μL normal saline (NS) was injected via the IJ line as a bolus injection (1–2 s). After 2-3 min, baseline blood pressure was noted and increasing concentrations of either phenylephrine, angiotensin II, acetylcholine, or sodium nitroprusside (Sigma, St. Louis, MO) each in an approximately 10 μL volume were injected in the IJ line and flushed with 40 μL NS. After each drug, when blood pressure returned to baseline (2-3 minutes), the line was washed with 50 μL NS for 3 min. The maximal change in blood pressure after each dose was reported. Mice were euthanized at the completion of the experiment.

## Pressure myography

Following blood pressure measurement and euthanasia, ascending aorta and left common carotid artery were excised and placed in physiologic saline solution (PSS) composed of 130 mM NaCl, 4.7 mM KCl, 1.6 mM $CaCl_2$, 1.18 mM $MgSO_4–7H_2O$, 1.17 mM $KH_2PO_4$, 14.8 mM $NaHCO_3$, 5.5 mM dextrose, and 0.026 mM Ethylenediaminetetraacetic acid (EDTA, pH 7.4) overnight at $4\,°C$. Vessels were cleaned of surrounding fat, mounted on a pressure arteriograph (Danish Myo Technology) and maintained in PSS at $37\,°C$. Vessels were visualized with an inverted microscope connected to a CCD camera and a computerized system, which allows continuous recording of vessel diameter. Intravascular pressure was increased from 0 to 175 mmHg by 25-mmHg increments, the vessel outer diameter was recorded at each step (12 s per step). The average of three measurements at each pressure was reported.

## Metabolic phenotyping

Male $Svep1^{+/+}$ and $Svep1^{-/-}$ mice were weighed and fasted for 5 hours prior to the metabolic challenge. For insulin tolerance tests (ITT), Humulin R (100 units/mL) was diluted 1:1000 in sterile PBS and injected intraperitoneally in mice at a dose of 0.75 units/kg. For glucose tolerance tests (GTT), a 20% glucose solution was prepared in sterile PBS and injected intraperitoneally at a final dose of 2 g/kg. Tail vein glucose measurements were collected at 15–30-min intervals using a glucometer. Mouse lean, fat, and total water mass were determined in male $Svep1^{+/+}$ and $Svep1^{-/-}$ mice fed HFD using EchoMRI (EchoMRI LLC). The EchoMRI was calibrated with canola oil. Measurements were gathered in duplicate for each mouse and averaged prior to analysis. Indirect calorimetry measurements were collected using the PhenoMaster System (TSE Systems) in collaboration with the Washington University Diabetes Research Center Diabetes Models

Phenotyping Core. Male $Svep1^{+/+}$ and $Svep1^{-/-}$ mice fed HFD were placed in individual chambers and acclimated for several hours prior to data collection. Mice were fed HFD throughout the data collection. The measurements occurred at room temperature during standard 12-hour light/12-hour dark cycles.

## Proteomic pulldown assays

Affinity based proteomics: Murine VSMCs were grown to confluence in serum-containing media and changed to serum-free media to generate enriched media. Recombinant, Myc-tagged SVEP1 was added to the media after two days of enrichment and incubated for 1 h. An aliquot was removed after incubation as the "input" fraction. Media was then added to a slurry of Pierce Anti-c-Myc Magnetic Beads + 0.05% Tween 20 and incubated for 30 min while rocking at $4\,°C$. The beads were then washed twice with $Ca^{2+}$ and $Mg^{2+}$-containing PBS (D8662, Sigma) + 0.05% Tween 20 and twice with $Ca^{2+}$ and $Mg^{2+}$-containing PBS before a final resuspension in PBS. An aliquot of the beads was reserved as the pulldown fraction for validation. Proximity based proteomics: recombinant SVEP1-Myc or SVEP1-mTID was added to enriched VSMC media with 500 μM exogenous biotin and 1 mM adenosine triphosphate (ATP). The samples were incubated for 4 h prior to dialysis. Protease arrest (G-Biosciences) was added and excess biotin was dialyzed using a 10 kDa molecular weight cut-off (MWKO) Slide-A-Lyzer™ dialysis Cassette (ThermoFisher Scientific) in buffered saline + 1 mM EDTA. The samples were then transferred to 10 kDa MWKO Vivasin column (Sartorius Stedim Biotech), centrifuged, resuspended in RIPA buffer, centrifuged, and added to pre-washed Pierce Streptavidin Magnetic Beads. The samples and beads were incubated for 1 hour at room temperature or overnight at $4\,°C$ and then washed with the following solutions: RIPA buffer, 1 M KCl, 0.1 M $Na_2CO_3$, 2 M urea in 10 mM Tris HCl (pH = 8.0), and PBS[42]. The beads were then resuspended in PBS for peptide preparation.

## Peptide preparation

The peptides were prepared using a previously described method for on-bead tryptic digestion[76]. The beads were washed four times with 1 mL of 50 mM ammonium bicarbonate buffer (pH = 8.0) (ABC). The washed beads were resuspended in 40 μL of ABC buffer containing 8 M urea. The protein disulfide bonds were reduced using 2 μL of 0.5 M TCEP and incubation for 60 min at $30\,°C$. The reduced proteins were alkylated using 4 μL of a 0.5 M solution of iodoacetamide with incubation for 30 min at RT in the dark. The urea was diluted to a concentration of 1.5 M by adding 167 μL of 50 mM ABC buffer. After addition of LysC (1mAU), the samples were incubated for 2 hours at $30\,°C$ in a Thermomixer with gyration at 750 rpm. Trypsin (1 μg) was added and the samples were incubated overnight at $30\,°C$ in the Thermomixer gyrating at 750 rpm. The peptides were transferred to a 1.5 mL tube, the beads were washed with 50 μL of ABC buffer and the transfer and wash volumes were combined. Any residual detergent was removed by ethyl acetate extraction[77]. The peptide samples were acidified with Trifluoroacetic acid (TFA) to a final concentration of 1% (vol/vol) TFA (pH <2.0). The pH was checked with pH paper. The peptides were desalted using two micro-tips (porous graphite carbon, BIOMETNT3CAR) (Glygen) on a Beckman robot (Biomek NX), as previously described[78]. The peptides were eluted with 60% (vol/vol) MeCN in 0.1% (vol/vol) TFA. After adding TFA to a final concentration of 5%, the peptides were dried in a Speed-Vac (Thermo Scientific, Model No. Savant DNA 120 concentrator). The peptides were dissolved in 20 μL of 1% (vol/vol) MeCN in water. An aliquot (10%) was removed for quantification using the Pierce Quantitative Fluorometric Peptide Assay kit (Thermo Scientific, Cat. No. 23290). The remainder was transferred to autosampler vials (Sun-Sri, Cat. No. 200046), dried and stored at $-80\,°C$. Peptides were also prepared after release of proteins from antibody beads. The beads were washed with 1 mL of 50 mM cold phosphate-buffered saline (pH 8.0) (PBS) followed by elution with 30 μL of SDS buffer (4% (wt/vol), 100 mM Tris-HCl pH 8.0). The protein

disulfide bonds were reduced using 100 mM DTT with heating to 95 °C for 10 min. Peptides were prepared as previously described using a modification[76,77] of the filter-aided sample preparation method (FASP)[79]. The samples were mixed with 200 μL of 100 mM Tris-HCL buffer, pH 8.5 containing 8 M urea (UA buffer). The samples were transferred to the top chamber of a 30,000 MWCO cutoff filtration unit (Millipore, part# MRCF0R030) and spun in a microcentrifuge at 14,000×*g* for 10 min. An additional 200 μL of UA buffer was added and the filter unit was spun at 14,000×*g* for 15 to 20 min. The cysteine residues were alkylated using 100 μL of 50 mM Iodoacetamide (Pierce, Ref. No. A39271) in UA buffer. Iodoacetamide in UA buffer was added to the top chamber of the filtration unit. The samples were gyrated at 550 rpm for 30 min in the dark at RT using a Thermomixer (Eppendorf). The filter was spun at 14,000×*g* for 15 min and the flow through discarded. Unreacted iodoacetamide was washed through the filter with two sequential additions of 200 μL of 100 mM Tris-HCl buffer, pH 8.5 containing 8 M urea and centrifugation at 14,000×*g* for 15 to 20 min after each buffer addition. The flow through was discarded after each buffer exchange-centrifugation cycle. The urea buffer was exchanged with digestion buffer (DB), 50 mM ammonium bicarbonate buffer, pH 8. Two sequential additions of DB (200 μL) with centrifugation after each addition to the top chamber was performed. The top filter units were transferred to a new collection tube and 100 μL DB containing 1mAU of LysC (Wako Chemicals, cat. no. 129-02541) was added and samples were incubated at 37 °C for 2 h. Trypsin (1 μg) (Promega, Cat. No. V5113) was added and samples were incubated overnight at 37 °C. The filters were spun at 14,000×*g* for 15 min to recover the peptides in the lower chamber. The filter was washed with 50 μL of 100 mM ABC buffer and the wash was combined with the peptides. Residual detergent was removed by ethyl acetate extraction[77,79]. After extraction, the peptides were dried in a Speedvac concentrator (Thermo Scientific, Savant DNA 120 Speedvac Concentrator) for 15 min. The dried peptides were dissolved in 1% (vol/vol) TFA and desalted using two microtips (porous graphite carbon, BIOMEKNT3CAR) (Glygen) on a Beckman robot (Biomek NX), as previously described[78]. The peptides were eluted with 60 μL of 60% (vol/vol) MeCN in 0.1% (vol/vol) TFA and dried in a Speed-Vac (Thermo Scientific, Model No. Savant DNA 120 concentrator) after adding TFA to 5% (vol/vol). The peptides were dissolved in 20 μL of 1% (vol/vol) MeCNA in water. An aliquot (10%) was removed for quantification using the Pierce Quantitative Fluorometric Peptide Assay kit (Thermo Scientific, Cat. No. 23290). The remaining peptides were transferred to autosampler vials (Sun-Sri, Cat. No. 200046), dried and stored at −80 °C.

## UPLC-timsTOF mass spectrometry

UPLC-timsTOF mass spectrometry was used for the affinity-based proteomic experiment and replicate. The peptides were analyzed using a nano-Elute chromatograph coupled online to a hybrid trapped ion mobility-quadrupole time of flight mass spectrometer (timsTOF Pro, Bruker Daltonics, Bremen Germany) with a modified nano-electrospray source (CaptiveSpray, Bruker Daltonics). The mass spectrometer was operated in parallel accumulation-serial fragmentation (PASEF) mode[80]. The samples in 1% (vol/vol) aqueous formic acid (FA) were loaded (2 μL) onto a 75 μm i.d. × 25 cm Aurora Series column with CSI emitter (Ionopticks) on a Bruker nano-ELUTE (Bruker Daltonics). The column temperature was set to 50 °C. The column was equilibrated using constant pressure (800 bar) with 8 column volumes of solvent A (0.1% (vol/vol) aqueous FA). Sample loading was performed at constant pressure (800 bar) at a volume of 1 x sample pick-up volume plus 2 μL. The peptides were eluted using the one column separation mode with a flow rate of 400nL/min and using solvents A and B (0.1% (vol/vol) FA/MeCN): solvent A containing 2% B increased to 15% B over 60 min, to 25% B over 30 min, to 35% B over 10 min, to 80% B over 10 min and constant 80% B for 10 min. The MS1 and MS2 spectra were recorded from m/z 100 to 1700. Suitable precursor ions for PASEF-MS/MS were selected in real time from TIMS-MS survey scans by a PASEF scheduling algorithm[80]. A polygon filter was applied to the m/z and ion mobility plane to select features most likely representing peptide precursors rather than singly charged background ions. The quadrupole isolation width was set to 2Th for m/z < 700 and 3Th for m/z > 700, and the collision energy was ramped stepwise as a function of increasing ion mobility: 52 eV for 0–19% of the ramp time; 47 eV from 19 to 38%; 42 eV from 38 to 57%; 37 eV from 57 to 76%; and 32 eV for the remainder. The TIMS elution voltage was calibrated linearly using the Agilent ESI-L Tuning Mix (m/z 622, 922, 1222).

## UPLC-Orbitrap mass spectrometry

UPLC-Orbitrap mass spectrometry was used for the proximity-based proteomic experiment and replicate. The samples in formic acid (1%) were loaded (2.5 μL) onto a 75 μm i.d. × 50 cm Acclaim PepMap 100 C18 RSLC column (Thermo-Fisher Scientific) on an EASY nanoLC (Thermo Fisher Scientific) at a constant pressure of 700 bar at 100% A (0.1%FA). Prior to sample loading the column was equilibrated to 100% A for a total of 11 μL at 700 bar pressure. Peptide chromatography was initiated with mobile phase A (1% FA) containing 2% B (100%ACN, 1%FA) for 5 min, then increased to 20% B over 100 min, to 32% B over 20 min, to 95% B over 1 minute and held at 95% B for 19 min, with a flow rate of 300 nL/minute. A lower flow rate (250 nL/minute), 95% B was held for 29 min for the replicate experiment. The data was acquired in data-dependent acquisition (DDA) mode. The full-scan mass spectra were acquired with the Orbitrap mass analyzer with a scan range of m/z = 325–1500 (350–1500) and a mass resolving power set to 70,000. Ten data-dependent high-energy collisional dissociations were performed with a mass resolving power set to 17,500, a fixed lower value of m/z 100, an isolation width of 2 Da, and a normalized collision energy setting of 27. The maximum injection time was 60 ms for parent-ion analysis and product-ion analysis. The target ions that were selected for MS/MS were dynamically excluded for 20 s. The automatic gain control (AGC) was set at a target value of 3e6 ions for full MS scans and 1e5 ions for MS2. Peptide ions with charge states of one or >8 were excluded for HCD acquisition.

## Identification of proteins

The data from the timsTOF Data mass spectrometer were converted to peak lists using DataAnalysis (version 5.2, Bruker Daltonics). The MS2 spectra with charges +2, +3 and +4 were analyzed using Mascot software[81](Matrix Science, London, UK; version 2.5.1). Mascot was set up to search against a UniProt (ver October 2013) database of mouse proteins (43,296 entries), assuming the digestion enzyme was trypsin with a maximum of 2 missed cleavages allowed. The searches were performed with a fragment ion mass tolerance of 50 ppm and a parent ion tolerance of 50ppm. Carbamidomethylation of cysteine was specified in Mascot as a fixed modification. Deamidation of asparagine, deamidation of glutamine, acetylation of protein N-terminus and oxidation of methionine were specified as variable modifications. Peptides and proteins were filtered at 1% false-discovery rate (FDR) by searching against a reversed protein sequence database. MS raw data acquired using a hybrid-quadrupole-Orbitrap mass spectrometer (Q-Exactive Plus, Thermo Fisher) were converted to peak lists using Proteome Discoverer (version 2.1.0.81, Thermo-Fischer Scientific). MS/MS spectra with charges greater than or equal to two were analyzed using Mascot search engine[81](Matrix Science, London, UK; version 2.7.0). Mascot was set up to search against a UniProt database of mouse (version October 2013, 43,296 entries), assuming the digestion enzyme was trypsin with a maximum of 4 missed cleavages allowed. The searches were performed with a fragment ion mass tolerance of 0.02 Da and a parent ion tolerance of 20 ppm. Carbamidomethylation of cysteine was specified in Mascot as a fixed modification. Deamidation of asparagine, formation of pyro-glutamic acid from N-terminal glutamine, acetylation of protein N-terminus and oxidation of methionine were specified as variable

modifications. Peptides and proteins were filtered at 1% false-discovery rate (FDR) by searching against a reversed protein sequence database.

## Mass spectrometry analysis

A cumulative binomial distribution was used to determine which proteins were enriched at a threshold of $P < 0.10$ in the samples containing SVEP1 compared to negative control samples. The probability of success on a single trial was set to the null hypothesis of 0.5. Proteins were considered "Hits" if they achieved a reproducibility criterion of $P < 0.10$ three controlled experiments and a fourth experiment that lacked an experimental negative control. A meta-analysis was performed on the three independent, controlled experiments using Fisher's method.

## Gene cloning

Full length PEAR1: Human *PEAR1* cDNA was obtained from a pDONR221 plasmid (HSCD00863115, DNASU plasmid repository) by PCR and cloned into a modified pCMV6 plasmid (OriGene, Rockville, MD) with a Myc and poly-histidine C-terminal tag. The empty pCMV6 plasmid was used as the empty vector control in experiments. SVEP1 miniTurbo fusion protein (SVEP1-mTID): MiniTurbo cDNA (the promiscuous biotin ligase[42]) was amplified by PCR from the V5-miniTurbo-NES-pCDNA3 plasmid (a gift from Alice Ting, Addgene, plasmid 107170) and cloned downstream of the murine *Svep1* sequence and upstream of the Myc and poly-histidine tag in the pCMV6 plasmid. PEAR1ECD-Bio-His: The plasmid used for the PEAR1 ecto-domain expression was pTT3-PEAR1-bio-His[36] (a gift from Gavin Wright, Addgene plasmid 51860). *Svep1*: cloning of mouse *Svep1* cDNA, protein expression, and purification was described in detail previously[4]. A DNA sequence coding for a biotinylation domain (LHHILDAQKMLWNHR, recognized by the BirA enzyme[82]) was inserted into the C-terminus region of the *Svep1* cDNA construct, upstream of the Myc tag, by standard molecular biology procedures. Briefly, all proteins were expressed in 293 F cells (Invitrogen) and grown in FreeStyle expression media. Plasmids were transfected with 3 μg/mL of vector DNA plus 9 μg/mL Polyethylenimine (PEI) (25 kDa linear PEI, Polysciences, Inc.) at a cell density of $2.5 \times 10^6$ cells/mL. For the PEAR1 ecto-domain and SVEP1 biotinylation, 0.3 μg/μL of secreted BirA-8his plasmid (a gift from Gavin Wright, Addgene plasmid 32408) was co-transfected and supplemented with 0.1 mM biotin. Proteins were purified in an NGC chromatographic system (BioRad Lab) with 5 mL Nuvia IMAC resin (BioRad Lab) and polished using a Superose 6 increase 10/300 column (GE LifeSciences) with PBS as a carrier buffer.

## Reporting summary

Further information on research design is available in the Nature Portfolio Reporting Summary linked to this article.

# Data availability

Data from the INTERVAL proteomics study were obtained from the European Genome-phenome Archive under Study ID EGAS00001002555. Data from the deCODE proteomics study were obtained from deCODE [https://download.decode.is/form/folder/proteomics]. UniProt (ver October 2013) database of mouse proteins was used in the mass spectrometry analysis. GTEx data used for the analyses described in this manuscript were obtained from the GTEx Portal [gtexportal.org] on 10/20/21. The raw mass spectrometry data generated in this study have been deposited in the MassIVE repository under accession number MSV000090134. Other data that support the findings of this study are provided in the Source Data file. Source data are provided with this paper.

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

## Acknowledgements

We thank Dr. Bruce Carter from Vanderbilt University for providing *Pear1⁻/⁻* mice. We thank the Washington University School of Medicine diabetes research center mouse phenotyping core for their help with the metabolic phenotyping. The expert technical assistance of Petra Erdmann-Gilmore, Dr. Yiling Mi and Rose Connors is gratefully acknowledged. The proteomic experiments were performed at the Washington University Proteomics Shared Resource (WU-PSR), R Reid Townsend MD, PhD (Director), Robert Sprung, PhD (Co-Director), Qiang Zhang, PhD (Co-Director). The WU-PSR is supported in part by the WU Institute of Clinical and Translational Sciences (NCATS UL1TR000448), the Mass Spectrometry Research Resource (NIGMS P41GM103422; R24GM136766) and the Siteman Comprehensive Cancer Center Support Grant (NCI P30CA091842). The Genotype-Tissue Expression (GTEx) Project was supported by the Common Fund of the Office of the Director of the National Institutes of Health, and by NCI, NHGRI, NHLBI, NIDA, NIMH, and NINDS. The data used for the analyses described in this manuscript were obtained from the GTEx Portal on 10/20/21. We thank Professor Adam Butterworth for assistance in obtaining access to the individual level data from the INTERVAL pQTL GWAS study. Participants in the INTERVAL randomized controlled trial were recruited with the active collaboration of NHS Blood and Transplant England (www.nhsbt.nhs.uk), which has supported field work and other elements of the trial. DNA extraction and genotyping was co-funded by the National Institute for Health Research (NIHR), the NIHR BioResource [http://bioresource.nihr.ac.uk] and the NIHR [Cambridge Biomedical Research Centre at the Cambridge University Hospitals NHS Foundation Trust]. The INTERVAL study was funded by NHSBT (11-01-GEN). The academic coordinating centre for INTERVAL was supported by core funding from: NIHR Blood and Transplant Research Unit in Donor Health and Genomics (NIHR BTRU-2014-10024), UK Medical Research Council (MR/L003120/1), British Heart Foundation (SP/09/002; RG/13/13/30194; RG/18/13/33946) and the NIHR [Cambridge Biomedical Research Centre at the Cambridge University Hospitals NHS Foundation Trust]. Proteomic assays were funded by the academic coordinating centre for INTERVAL and MRL, Merck & Co., Inc. A complete list of the investigators and contributors to the INTERVAL trial is provided elsewhere[83]. The academic coordinating centre would like to thank blood donor centre staff and blood donors for participating in the INTERVAL trial. This work was supported by Health Data Research UK, which is funded by the UK Medical Research Council, Engineering and Physical Sciences Research Council, Economic and Social Research Council, Department of Health and Social Care (England), Chief Scientist Office of the Scottish Government Health and Social Care Directorates, Health and Social Care Research and Development Division (Welsh Government), Public Health Agency (Northern Ireland), British Heart Foundation and Wellcome. The views expressed are those of the authors and not necessarily those of the NHS, the NIHR or the Department of Health and Social Care. This work was supported in part by grants from the National Institutes of Health (NIH) to JSE (T32GM007200, T32HL134635, and F30HL152521), CMH (K08HL135400), NOS (R01HL159171, R01HL131961, UM1HG008853, and P01HL151328), by the Longer Life Foundation: A RGA/Washington University Collaboration (LLF 2021-007 to NOS), by the Missouri ACC and the Missouri ACC Foundation (NOS), by the Foundation for Barnes-Jewish Hospital (NOS), and by the Diabetes Research Center at Washington University under NIH award number P30DK020579. Proteomic assays were funded in part by the Washington University Institute of Clinical and Translational Science under NIH award number UL1TR002345. The content is solely the responsibility of the authors and does not necessarily represent the official views of the National Institutes of Health.

## Author contributions

J.S.E. and N.O.S. designed the study. J.S.E., U.P., K.J.A., V.P., K.S., I.-H.J., P.C.L., K.H.B., C.M.H., and A.A. performed the experiments. J.S.E., U.P., K.J.A., V.P., P.C.L., R.P.M., C.M.H., A.A., J.D.P., and N.O.S. designed and interpreted the experiments. A.A., K.S., and J.M.A. generated critical reagents. C.J.K. performed multi-omics and MR analyses. J.S.E. and N.O.S. wrote the manuscript. N.O.S. acquired funds. All authors reviewed and provided critical editing and approved of the manuscript.

## Competing interests

N.O.S. has received investigator-initiated research funds from Regeneron Pharmaceuticals unrelated to the content of this study. J.S.E., I.H.J., A.A., and N.O.S. are co-inventors on a patent application filed by Washington University focused on SVEP1 and PEAR1. The other authors declare no competing interests.
