## [Peer Review File · Nature Communications]

SVEP1 is an endogenous ligand for the orphan receptor PEAR1REVIEWER COMMENTS

Reviewer #1 (Remarks to the Author):

In this investigation, Elenbaas and colleagues utilize a broad range of methodological approaches including multiomic (genomic/proteomic) characterization, molecular signaling studies, as well as cellular and animal models to provide novel evidence that SVEP1 is a ligand for PEAR1, and that the interaction of these proteins have implications for several platelet phenotypes and coronary artery disease risk. Recent high-profile investigations have shown that both of these genes/proteins associate with several diseases and clinically important phenotypes; however, relatively little is known regarding the mechanistic underpinnings regarding these associations. This is a well-written manuscript that is fairly easy to read despite the density of the data described. The major results of this investigation are novel and are likely to facilitate future investigations of PEAR1 and SVEP1 with regards to human health.

While this investigation has many strengths, there are some weaknesses that I ask the authors to consider below. Importantly, more methodological details seem warranted for the derivation of the genetic instruments used in Mendelian Randomization studies as well as association studies linking these instruments to platelet phenotypes and CAD risk. In comparison to the methodological details provided for the molecular signaling and animal model studies, the methods provided for the human studies seem relatively scant. Additional comments/suggestions are provided below.

1) In the author's initial proteomic analysis, they observed that their sentinel SNP (rs145662369) did not influence SVEP1 expression in GTex but was in perfect LD with rs147639000, a SNP that also does seem to impact expression according to GTex but associates with SVEP1 plasma levels in the INTERVAL Study. Have the authors performed conditional analyses to see if rs147639000 is driving most of the signal in this locus (i.e. most of the linked SNPs shown in Supplementary Figure 1a)? Furthermore, many prior GWAS and other genetic studies of PEAR1 have focused on intronic variant rs12041331 (or rs12566888 which is in high LD with rs12041331). Given the fact that the sentinel SNP in the current investigation is only approximately 5kb away from these variants have the authors considered running conditional analyses to test if these nearby SNPs are impacting the association between the current SNPs and SVEP1 levels. While the R^2 between these variants is low (due to differences in minor allele frequency, the D' =1 suggesting they reside in the same LD block).

2) It would be helpful if more information was available in the Methods section regarding derivation of the genetic instruments of PEAR1 and SVEP1 levels in generated from deCODE and then subsequently implemented in INTERVAL. What criteria was used to include individual SNPs into the instrument? How many SNPs were included in each instrument and what was the approximate percentage of variation in plasma levels of each protein that were captured by these instruments?

3) The authors hypothesize that genetic variation that influences PEAR1 expression would also impact SVEP1 expression. As noted, however, there was little evidence to suggest that rs147639000 meaningfully influenced PEAR1 levels? This might be expected given that rs147639000 is a missense mutation and may not mediate any potential effect by influencing mRNA/protein levels. In that case, however, can the authors speculate how this variant may be impacting SVEP1 levels? Any experimental evidence showing the effect of PEAR1 rs147639000 on SVEP1 (e.g. reduced binding kinetics, altered co-immunoprecipitation, etc.) would greatly strengthen the genomic data presented in the opening of the results section.

Related to this point, on page 5 line 103 the authors mention that this variant influences PEAR1 expression in vascular endothelial cells and cite a publication by Vandenbrielle et al. Perhaps this reviewer missed it but I did not see data suggesting that this variant influenced expression of PEAR1 in this paper. Please double check this reference and/or reword this sentence.

4) The authors nicely showed that SVEP1 led to HUVEC adhesion in a concentration-specific manner (Figure 3I). Given the evidence presented in this investigation that SVEP1 is a ligand for PEAR1, one

might expect that lack of PEAR1 in HUVECs would result in decreased adhesion through abrogation of the interaction with SVEP1. Have the authors considered performing adhesion experiments after siRNA-mediated knockdown of PEAR1 in HUVECs?

5) When characterizing the effect of SVEP1 on platelet function, the authors use change in whole blood platelet count as a marker of platelet aggregation. This is a relatively coarse way to evaluate this. A more specific method would've been to use some type of turbidimetric method such as platelet aggregometry (in whole blood or PRP). This would've been especially powerful given the number of PEAR1 studies devoted to platelet aggregation as well as the recent GWAS identifying SVEP1 variants as a major determinant of platelet aggregation, which use such methods.

6) More information regarding the MR analyses of human platelet phenotypes and CAD is warranted in both the methods and results sections. For example, what were the clinical characteristics of participants and sample sizes for each of these analyses? Were both healthy individuals and patients with CAD used in the PLT and MPV analyses...or were those analyses limited only to healthy people in INTERVAL? Were the exact same genetic instruments used/available among the 3 cohorts (INTERVAL, UK Biobank, and CARIDoGRAMPlusC4D)? What were the effect sizes (e.g. betas, OR/HR, etc.)? How was CAD defined? I realize that some of this information may be available from INTERVAL, UK Biobank, and CARIDoGRAMPlusC4D. However, these are independent analyses using a distinct set of genetic markers for the MR instruments and readers shouldn't be expected to track down and parse a lot of this information. As currently written, there isn't a lot of information to go off.

Minor comments

1) In Figure 2A, could the authors please include the nM range of SVEP1 next to the blue lines or in some way identify the concentrations?

2) When transfecting PEAR1 into 293T cell, the authors noted constitutive activation of AKT. Can the authors surmise what caused this constitutive activation? It would've been reassuring to see the same pattern of AKT activation in these cells after exposure to SVEP1, PEAR1 pAb, etc...as I assume the authors were trying recapitulate.

3) On page 11, lines 269-271, the authors should mention that these data are presented in Figure S4 B-D.

4) Figure 5E is confusing to me. First, on the top portion of the figure, there appears to be 4 lines for 3 experimental groups (based on the legend to the right). Furthermore, it appears that platelets adhered to BSA to the same extent as they adhered to SVEP1 and fibrinogen. Finally, the Y-axis seems inconsistent with the +/- thrombin...both are around 1? Perhaps I am reading these data incorrectly but please check the data included as well as the unit of measure on the Y-axis.

Reviewer #2 (Remarks to the Author):

The following review regards the BLI experiment depicted in Figure 2A.

The authors' are utilizing BLI to show a direct interaction between PEAR1 and SVEP1. Based on the Methods section, I believe the experiment was performed adequately. However I have a couple comments:

- BLI measures changes in the wavelength of light in nanometers, which is an indirect measurement

of protein-binding. The y-axis should be annotated as such ("shift" instead of the current "protein binding").

- This experiment lacks a negative control. The minimum negative control for BLI is to test the association of your protein of interest to an unloaded sensor. The authors describe this experiment in the Methods section, but to my knowledge do not include it in their manuscript. Showing little to no association of SVEP1 to the biosensor at the highest concentration used for their experiments (80 nM) is critical. Ideally, one would want to show little to no association to a mutant PEAR1/SVEP1. I do not believe the authors identified where the two proteins interact, so I understand this may not be feasible with their current knowledge of the two proteins. Instead, I would suggest performing a BLI experiment with loading of a separate protein that should not interact with SVEP1, such as BSA. If SVEP1 shows little to no interaction with this protein, it would provide decent evidence that the interaction to PEAR1 is specific.

Reviewer #3 (Remarks to the Author):

This study by Elenbaas and colleagues proposes a role for a CVD-associated gene (SVEP1) in PEAR1 stimulation and platelet activation. This manuscript follows a series of high impact publications related to the SVEP1 gene from this outstanding group, where they showed a link to CAD and atherosclerosis. A variety of elegant approaches are used, including human genetics, molecular biology, and mouse studies. This reviewer finds the possibility of a trans-pQTL effect very interesting, and the MR studies compelling. The signaling studies are also well done, especially with the controls related to SFK and clathrin inhibitors. The phenotyping in the in vivo studies appears robust, especially going so far as to do difficult studies such as arterial catheterization. It is somewhat confusing as to why the mouse models used could recapitulate the human link with platelet phenotypes, but not the metabolic or vascular phenotypes. Having said that, the paper is very nice and this reviewer is curious about the following questions, which hopefully could strengthen an already very interesting story.

1. It is fascinating that variants which alter PEAR1 levels are inversely associated with SVEP1 levels in the plasma. Figures 1F and 1G are beautifully executed, and support the sequestration hypothesis. However, the main variant of interest in this manuscript (D343N via rs145665369 with which it is in perfect LD) does not alter PEAR1 levels. So if the risk allele does not alter PEAR1 level, it is difficult to state that the altered SVEP1 level is because there is more/less PEAR1 around to sequester it. What mechanism do the authors propose causes this trans-PQTL effect, and is there evidence to support the hypothesis? I realize that crystallography and related experiments are probably outside the scope of this article. However, some of the binding assays from Fig2 could be repeated with protein harboring the missense mutation, to prove that binding kinetics are altered. A final point related to this query is that if a post-transcriptional mechanism is being proposed (as elegantly shown with the proteomics data in Fig 1E), it would be really neat to show that there is no such trans associations at the mRNA level.

2. In Fig 3A, why does SVEP1 induce so much less phosphorylation than the activating PEAR1 Ab? Were other potential PEAR1 ligands compared to be sure that SVEP1's effects are specific and robust compared to other potential receptor ligands?

3. I don't fully understand what Fig 3I shows us. Were different doses of fibronectin tested? Were other factors that cause HUVEC adherence tested? Did the binding decrease when PEAR1-deficient HUVECs were studied?

4. I also don't understand what the lamellipodia experiments in Fig 4A/B teach us, from a functional perspective. Can the authors explain what biological insights this provides, especially for the later studies related to platelet activation?

5. In Fig 4E, the first two columns say DMSO, but show different results. Was this a typo? What were the authors trying to show here?

6. In the SVEP1 KO studies from Fig 5 and Fig S4, the authors call this a 'diet-induced diabetes' model, but seem to have only fed western diet in their studies. Were these mice on a different background (e.g. db/db, etc) or was this not a classic diabetes model? Furthermore, the authors state

there was no metabolic impact of loss of this gene. However, the introduction also states that genetic variation in SVEP1 is associated with diabetes. Do the authors have an idea of why the mouse model doesn't reflect the human genetics?

7. Along the same lines, the authors state that BP is linked to SVEP1 in humans, but no difference was reported in Fig S5 – can the authors please comment?

8. In Fig 6, why is CVD used as the phenotype, rather than one more specifically related to thrombosis/platelet activation. Especially because the link between SVEP1 and CAD has previously been described (Ref 4), it would be good to show some link to a clinically relevant clotting disorder (e.g. DVT, PE, etc). I think this could be important given the risk of pleiotropy with a phenotype as broad as CVD, especially for a factor that is linked to a number of cardiometabolic phenotypes in humans.

Minor:

- In the introduction, the text describes an association for a variant with CAD risk as being shown in Fig 1A. However, that figure does not describe such an association and instead shows the structure of the gene. This reviewer suggests rephrasing of these sentences.

- Are the blue circle and square in the figure legend for Fig 5J extraneous?

- The color scheme of Figs 5K and 5L are somewhat difficult for this reviewer to resolve and might be enhanced with different colors for clarity.

Reviewer #4 (Remarks to the Author):

I have no real comment to make on all the expression studies, interaction studies, platelet signalling studies, PEAR1 and Akt phosphorylation studies in the various cells, not do I find shortcomings in the adhesion studies and interaction studies between PEAR1 and SVEP1. Except for the few minor comments made below, I agree with every statement made in this paper. I want to congratulate the authors for such a comprehensive study, which I rarely met during a first round of submission.

I just have 3 small, minor things:

1. The authors mention in the discussion that FcEpsilonR1alpha is not a physiological ligand because it does not activate platelets. That is only correct for the monomeric recombinant FcEpsilonR1alpha chain, but its pentameric equivalent (as it is present on immune cells) does trigger potent platelet activation via PEAR1. Whereas it is most probably correct that FcEpsilonR1alpha is not the prime ligand for PEAR1, the interaction is not nonproductive and can lead to human platelet activation, be it much less to murine platelet activation.

2. This study would benefit from some immunohistochemical staining of SVEP1 in basement membrane or in cross-sections of arterial tissue, to illustrate its localisation in subendothelial space vs media and/or adventitia (this may be somewhat redundant with pictures published before, but the emphasis should be put on the compartmental analysis, as part of the activation mechanism of endothelial cells and/or platelets, during injury, atherosclerosis).

3. Whereas the strength and specificity of Mendelian Randomisation is well discussed (Discussion), this has not been done for the INTERVAL approach. It would be informative in the Supplement or the Methods to read a few lines on the specificity and strength of this methodology. Such would facilitate interpreting strength and analysis in Figure 1.

Response to Reviewer's Comments

We thank the Reviewers for their extremely thorough and thoughtful evaluation of this manuscript along with their helpful suggestions. In response to the Reviewer's comments, we have revised the text, performed additional supplementary analyses, conducted additional experiments, and incorporated new results into the revised manuscript. We detail our point-by-point response to the Reviewers' inquiries and suggestions below.

Reviewer #1

In this investigation, Elenbaas and colleagues utilize a broad range of methodological approaches including multiomic (genomic/proteomic) characterization, molecular signaling studies, as well as cellular and animal models to provide novel evidence that SVEP1 is a ligand for PEAR1, and that the interaction of these proteins have implications for several platelet phenotypes and coronary artery disease risk. Recent high-profile investigations have shown that both of these genes/proteins associate with several diseases and clinically important phenotypes; however, relatively little is known regarding the mechanistic underpinnings regarding these associations. This is a well-written manuscript that is fairly easy to read despite the density of the data described. The major results of this investigation are novel and are likely to facilitate future investigations of PEAR1 and SVEP1 with regards to human health. While this investigation has many strengths, there are some weaknesses that I ask the authors to consider below. Importantly, more methodological details seem warranted for the derivation of the genetic instruments used in Mendelian Randomization studies as well as association studies linking these instruments to platelet phenotypes and CAD risk. In comparison to the methodological details provided for the molecular signaling and animal model studies, the methods provided for the human studies seem relatively scant. Additional comments/suggestions are provided below.

Author reply: We thank Reviewer 1 for their comprehensive review of our manuscript and for these comments.

1) In the author's initial proteomic analysis, they observed that their sentinel SNP (rs145662369) did not influence SVEP1 expression in GTex but was in perfect LD with rs147639000, a SNP that also does seem to impact expression according to GTex but associates with SVEP1 plasma levels in the INTERVAL Study. Have the authors performed conditional analyses to see if rs147639000 is driving most of the signal in this locus (i.e. most of the linked SNPs shown in Supplementary Figure 1a)?

Author reply: We thank the reviewer for this suggestion. In conditional analysis using the INTERVAL study data it does appear that rs14763900 is driving most of the signal in the PEAR1 locus. There may be a second independent signal in the locus; however, this does not reach genome-wide significance in INTERVAL. It is possible this would be validated as a second independent signal in deCODE (N~30,000 in deCODE vs N~3,000 in INTERVAL) but unfortunately individual-level proteomic and genetic data are not available for the deCODE study.

Reviewer Figure 1. Original SVEP1 pQTL association results in the *PEAR1* locus.

Reviewer Figure 2. SVEP1 pQTL association results in the *PEAR1* locus when adjusting for rs147639000 as a covariate.

Furthermore, many prior GWAS and other genetic studies of *PEAR1* have focused on intronic variant rs12041331 (or rs12566888 which is in high LD with rs12041331). Given the fact that the sentinel SNP in the current investigation is only approximately 5kb away from these variants have the authors considered running conditional analyses to test if these nearby SNPs are impacting the association between the current SNPs and SVEP1 levels. While the R^2 between these variants is low (due to differences in minor allele frequency, the D' =1 suggesting they reside in the same LD block).

Author reply: In the INTERVAL study, rs12041331 is not associated with SVEP1 levels in unadjusted (P-value for SVEP1 level = 0.16) or conditional analysis using *PEAR1* D343N (rs14763900) as a covariate (P-value for SVEP1 level = 0.1), suggesting that it is not affecting the association between the *PEAR1* SNPs and SVEP1 levels. Although rs12041331 is a pQTL for *PEAR1*, we suspect this apparent incongruity may be related to differential effects of the SNPs on different cell-types, perhaps through influencing the appearance of *PEAR1* in the plasma from tissues that do not express SVEP1, for example. Unfortunately, we are not currently able to test this hypothesis in a rigorous way.

2) It would be helpful if more information was available in the Methods section regarding derivation of the genetic instruments of *PEAR1* and SVEP1 levels in generated from deCODE and then subsequently implemented in INTERVAL. What criteria was used to include individual SNPs into the instrument? How many SNPs were included in each instrument and what was the approximate percentage of variation in plasma levels of each protein that were captured by these instruments?

Author reply: We thank the Reviewer for these suggestions. In the Methods, we specify the LD threshold ($r^2 \leq 0.2$), significance threshold (P-value for respective plasma protein concentration $\leq 5 \times 10^{-8}$), and genomic location (within 250kb of gene) for including SNPs into the instruments

for SVEP1 and PEAR1 that were generated using the deCODE summary statistics. We have added included additional detail in the Methods section regarding the number of SNPs in each instrument. We believe this is sufficient for others to recreate the instruments used in our analyses but we are happy to add additional details if the Reviewer has specific suggestions that are not currently included. Because individual level data are not available in deCODE, we are unable to estimate the percentage of variation in plasma levels of each protein that were captured by these instruments (this would be the most relevant piece of data to include in the manuscript). For the Reviewer's benefit, we used the genomic-relatedness-based restricted maximum-likelihood (GREML) approach as implemented in GCTA¹ to estimate the variance explained by the deCODE-derived instruments in INTERVAL. The SVEP1 instrument created in deCODE explains 7.4% of the variance in SVEP1 levels in INTERVAL, while the PEAR1 instrument explains 2.4% of the variance in PEAR1 levels.

3) The authors hypothesize that genetic variation that influences PEAR1 expression would also impact SVEP1 expression. As noted, however, there was little evidence to suggest that rs147639000 meaningfully influenced PEAR1 levels? This might be expected given that rs147639000 is a missense mutation and may not mediate any potential effect by influencing mRNA/protein levels. In that case, however, can the authors speculate how this variant may be impacting SVEP1 levels? Any experimental evidence showing the effect of PEAR1 rs147639000 on SVEP1 (e.g. reduced binding kinetics, altered co-immunoprecipitation, etc.) would greatly strengthen the genomic data presented in the opening of the results section.

Author reply: We agree with the Reviewer that demonstrating a functional effect of the rs147639000 (PEAR1 p.D343N) variant would strengthen this manuscript. Based on the findings contained within this manuscript, we designed two experiments to test hypotheses related to this question. First, we tested if cells expressing PEAR1 D343N may exhibit less AKT activation in response to SVEP1 than cells expressing WT PEAR1. Unfortunately, expression of either PEAR1 variant resulted in saturation of AKT signaling in 293T cells, obscuring the interpretability of the data (Reviewer Figure 3).

Reviewer Figure 3. PEAR1 and PEAR1D343N over-expression in 293T cells results in saturation of phospho-AKT.

Next, we tested if PEAR1 D343N has reduced affinity for SVEP1 using BLI with SVEP1-loaded biosensors. It appears that PEAR1 D343N may have lower affinity to SVEP1 compared to WT PEAR1 (1.76nM vs. 0.708nM, Figure S2B,C). Although this apparent difference is directionally consistent with the hypothesis that D343N results in increased plasma SVEP1 via decreased receptor affinity, we believe that inherent differences between the in vitro assay and in vivo human physiology (the BLI measures the affinity between immobilized SVEP1 and the soluble extracellular domain of PEAR1, for example) preclude a definitive conclusion that the trans

¹ Yang J, Lee SH, Goddard ME, Visscher PM. GCTA: a tool for genome-wide complex trait analysis. *Am J Hum Genet.* 2011 Jan 7;88(1):76-82.

pQTL effect is a result of altered affinity. As such, we have qualified our interpretation of these results in the manuscript. Unfortunately, the literature regarding mechanisms of PEAR1 signaling is limited (in large part due to the previous lack of a known endogenous ligand). We hope our manuscript will enable future studies that may provide a deeper understanding into PEAR1 signaling and the molecular mechanisms of rs147639000.

Related to this point, on page 5 line 103 the authors mention that this variant influences PEAR1 expression in vascular endothelial cells and cite a publication by Vandenbrielle et al. Perhaps this reviewer missed it but I did not see data suggesting that this variant influenced expression of PEAR1 in this paper. Please double check this reference and/or reword this sentence.

Author reply: We apologize for the prior wording which was unclear. The sentence has been reworded for clarity and now reads: "Given the impact of the PEAR1 D343N variant on plasma SVEP1 concentration and because PEAR1 is expressed on vascular endothelial cells, we hypothesized that PEAR1 binds and sequesters circulating SVEP1 from human plasma."

4) The authors nicely showed that SVEP1 led to HUVEC adhesion in a concentration-specific manner (Figure 3I). Given the evidence presented in this investigation that SVEP1 is a ligand for PEAR1, one might expect that lack of PEAR1 in HUVECs would result in decreased adhesion through abrogation of the interaction with SVEP1. Have the authors considered performing adhesion experiments after siRNA-mediated knockdown of PEAR1 in HUVECs?

Author reply: We thank the Reviewer for this excellent point. Reviewer 3 raised similar concerns about Figure 3I. We have been unsuccessful in reducing PEAR1 levels in HUVECs and therefore cannot test whether SVEP1-induced HUVEC adhesion is dependent on PEAR1. Although we avoided making this claim in the original manuscript, we are concerned that the context may lead to inappropriate inferences. For this reason, we have decided to remove Figure 3I.

5) When characterizing the effect of SVEP1 on platelet function, the authors use change in whole blood platelet count as a marker of platelet aggregation. This is a relatively coarse way to evaluate this. A more specific method would've been to use some type of turbidimetric method such as platelet aggregometry (in whole blood or PRP). This would've been especially powerful given the number of PEAR1 studies devoted to platelet aggregation as well as the recent GWAS identifying SVEP1 variants as a major determinant of platelet aggregation, which use such methods.

Author reply: We agree with the Reviewer and now present platelet aggregometry from PRP exposed to SVEP1 in Figure 5F. A corresponding methods section has also been added.

6) More information regarding the MR analyses of human platelet phenotypes and CAD is warranted in both the methods and results sections. For example, what were the clinical characteristics of participants and sample sizes for each of these analyses? Were both healthy individuals and patients with CAD used in the PLT and MPV analyses...or were those analyses limited only to healthy people in INTERVAL? Were the exact same genetic instruments used/available among the 3 cohorts (INTERVAL, UK Biobank, and CARIDoGRAMPlusC4D)? What were the effect sizes (e.g. betas, OR/HR, etc.)? How was CAD defined? I realize that some of this information may be available from INTERVAL, UK Biobank, and CARIDoGRAMPlusC4D. However, these are independent analyses using a distinct set of genetic markers for the MR instruments and readers shouldn't be expected to track down and parse a lot of this information. As currently written, there isn't a lot of information to go off.

Author reply: The Reviewer raises important points surrounding the definition of the outcomes tested in our two sample Mendelian Randomization (MR) analyses. The CAD, PLT, and MVP outcomes for our MR were extracted from published summary statistics of prior large-scale genome-wide association study (GWAS) meta-analyses encompassing many sub-studies. For example, CARDIoGRAMPlusC4D is a meta-analysis of 48 individual sub studies each of which have a study specific definition of CAD; similarly, the PLT and MVP outcomes were obtained from prior meta-analysis of GWAS from more than 25 sub studies each of which have specific details on participant demographics. As such, we do not feel a short summary would be adequate in providing a sufficient level of detail and instead include references to those studies in the Methods for readers who seek additional details. The same genetic instrument (PEAR1 and SVEP1 levels in deCODE) were used as the exposure for all four outcomes (protein levels in INTERVAL, CAD, PLT, and MVP). The beta and p-value for all MR analyses are reported in the figure legends.

Minor comments

1) In Figure 2A, could the authors please include the nM range of SVEP1 next to the blue lines or in some way identify the concentrations?

Author reply: We thank the Reviewer for this suggestion and now include the concentration of SVEP1 used in the BLI assay within the Figure.

2) When transfecting PEAR1 into 293T cell, the authors noted constitutive activation of AKT. Can the authors surmise what caused this constitutive activation? It would've been reassuring to see the same pattern of AKT activation in these cells after exposure to SVEP1, PEAR1 pAb, etc...as I assume the authors were trying recapitulate.

Author reply: The Reviewer is correct that the intent of this experiment was to test whether reconstitution of PEAR1 in cells lacking PEAR1 would enable a response to SVEP1 and PEAR1 pAb. There are several potential mechanisms to explain why over expression may result in constitutive activation of PEAR1. One possibility is that over expression of PEAR1 may overwhelm the post-translational machinery that may be required to properly regulate PEAR1. Another possibility is that a certain stoichiometry with an unknown inhibitor may be necessary for regulating PEAR1 and this balance could be disrupted when PEAR1 is overexpressed. Unfortunately, PEAR1 signaling is poorly understood, which limits our ability to speculate in the manuscript with any confidence.

3) On page 11, lines 269-271, the authors should mention that these data are presented in Figure S4 B-D.

Author reply: We thank the Reviewer for noting this omission and have updated the text accordingly.

4) Figure 5E is confusing to me. First, on the top portion of the figure, there appears to be 4 lines for 3 experimental groups (based on the legend to the right). Furthermore, it appears that platelets adhered to BSA to the same extent as they adhered to SVEP1 and fibrinogen. Finally, the Y-axis seems inconsistent with the +/- thrombin...both are around 1? Perhaps I am reading these data incorrectly but please check the data included as well as the unit of measure on the Y-axis.

Author reply: We apologize for the confusion and have edited the Figure axis to address these concerns.

Reviewer #2

The following review regards the BLI experiment depicted in Figure 2A.

Author reply: We thank Reviewer 2 for their review of our BLI experiment.

The authors' are utilizing BLI to show a direct interaction between PEAR1 and SVEP1. Based on the Methods section, I believe the experiment was performed adequately. However I have a couple comments:

- BLI measures changes in the wavelength of light in nanometers, which is an indirect measurement of protein-binding. The y-axis should be annotated as such ("shift" instead of the current "protein binding").

Author reply: We thank the Reviewer for this suggestion; we have now annotated the y-axis to read "shift".

- This experiment lacks a negative control. The minimum negative control for BLI is to test the association of your protein of interest to an unloaded sensor. The authors describe this experiment in the Methods section, but to my knowledge do not include it in their manuscript. Showing little to no association of SVEP1 to the biosensor at the highest concentration used for their experiments (80 nM) is critical. Ideally, one would want to show little to no association to a mutant PEAR1/SVEP1. I do not believe the authors identified where the two proteins interact, so I understand this may not be feasible with their current knowledge of the two proteins. Instead, I would suggest performing a BLI experiment with loading of a separate protein that should not interact with SVEP1, such as BSA. If SVEP1 shows little to no interaction with this protein, it would provide decent evidence that the interaction to PEAR1 is specific.

Author reply: We thank the Reviewer for highlighting the utility of presenting negative control data. In Supplemental Figure S2 we now show there is no association of SVEP1 to the unloaded biosensor. We agree that repeating these experiments with binding-null proteins would be strengthen these experiments. This specific line of investigation is a work in progress and we have not yet identified the minimum-binding region of the proteins to enable this experiment. As recommended by the Reviewer, we have also added BLI data for PEAR1 to BSA binding (Supplemental Figure S2).

Reviewer #3

This study by Elenbaas and colleagues proposes a role for a CVD-associated gene (SVEP1) in PEAR1 stimulation and platelet activation. This manuscript follows a series of high impact publications related to the SVEP1 gene from this outstanding group, where they showed a link to CAD and atherosclerosis. A variety of elegant approaches are used, including human genetics, molecular biology, and mouse studies. This reviewer finds the possibility of a trans-pQTL effect very interesting, and the MR studies compelling. The signaling studies are also well done, especially with the controls related to SFK and clathrin inhibitors. The phenotyping in the in vivo studies appears robust, especially going so far as to do difficult studies such as arterial catheterization. It is somewhat confusing as to why the mouse models used could recapitulate the human link with platelet phenotypes, but not the metabolic or vascular phenotypes. Having said that, the paper is very nice and this reviewer is curious about the following questions, which hopefully could strengthen an already very interesting story.

Author reply: We thank Reviewer 3 for their comprehensive review of our manuscript and for these comments.

1. It is fascinating that variants which alter PEAR1 levels are inversely associated with SVEP1 levels in the plasma. Figures 1F and 1G are beautifully executed, and support the sequestration hypothesis. However, the main variant of interest in this manuscript (D343N via rs145665369 with which it is in perfect LD) does not alter PEAR1 levels. So if the risk allele does not alter PEAR1 level, it is difficult to state that the altered SVEP1 level is because there is more/less PEAR1 around to sequester it. What mechanism do the authors propose causes this trans-PQTL effect, and is there evidence to support the hypothesis? I realize that crystallography and related experiments are probably outside the scope of this article. However, some of the binding assays from Fig2 could be repeated with protein harboring the missense mutation, to prove that binding kinetics are altered. A final point related to this query is that if a post-transcriptional mechanism is being proposed (as elegantly shown with the proteomics data in Fig 1E), it would be really neat to show that there is no such trans associations at the mRNA level.

Author reply: We thank the Reviewer for these comments. In a new Figure S1C, we demonstrate the lack of a trans association between rs147639000 and *SVEP1* transcription, as recommended by the Reviewer. In addition, we performed the binding assay experiment suggested by the Reviewer and found that PEAR1 D343N has an apparent 2.5-fold decrease in affinity for SVEP1. However, as we discuss above in our response to Question 3 from Reviewer 1, we are unable to definitively conclude if this is responsible for the observed increase in plasma SVEP1.

2. In Fig 3A, why does SVEP1 induce so much less phosphorylation than the activating PEAR1 Ab? Were other potential PEAR1 ligands compared to be sure that SVEP1's effects are specific and robust compared to other potential receptor ligands?

Author reply: We thank the Reviewer for this interesting observation. We believe this difference in phosphorylation is primarily due to differences in the kinetics of how cells interact with a soluble antibody compared to an immobilized ECM protein. Upon its addition to media, an antibody can quickly coat the receptors on the cells, resulting in rapid and potent signaling. In contrast, interaction with SVEP1 is much slower, since it involves cells adhering to the matrix and spreading its membrane to form new connections. Anecdotally, the experiments with longer incubation times generally resulted in similar levels of activation between SVEP1 and PEAR1 pAb, whereas experiments with shorter incubation times resulted in disparate levels of activation. Given their markedly different kinetics, it is difficult to compare signaling intensity between SVEP1 and PEAR1 pAB. The bivalent nature of antibodies also confounds the

analysis. We are not aware of any additional PEAR1 pseudoligands that would overcome these limitations.

3. I don't fully understand what Fig 3I shows us. Were different doses of fibronectin tested? Were other factors that cause HUVEC adherence tested? Did the binding decrease when PEAR1-deficient HUVECs were studied?

Author reply: We thank the Reviewer for raising this important point; similar concerns were raised by Reviewer 1. We decided to remove Figure 3I, since it does not meaningfully add to the interpretation of any results and may result in inappropriate inferences.

4. I also don't understand what the lamellipodia experiments in Fig 4A/B teach us, from a functional perspective. Can the authors explain what biological insights this provides, especially for the later studies related to platelet activation?

Author reply: We agree with the Reviewer that the results of Figure 4A and B are not particularly insightful in isolation. The SVEP1/PEAR1 interaction hypothesis was based on a unique approach; therefore, we sought to interrogate their interaction using as many complimentary approaches as possible. We believe this strategy greatly adds to the rigor of this study, but it does diminish the biological insights of individual experiments, as noted by the Reviewer. We believe the experiments in Figure 4A,B and S3C,D provide additional evidence of the interaction between SVEP1 and PEAR1, validate prior literature regarding the signaling mechanisms of PEAR1, and add spatial insight into how SVEP1 activates AKT by signaling through PEAR1.

5. In Fig 4E, the first two columns say DMSO, but show different results. Was this a typo? What were the authors trying to show here?

Author reply: We apologize for the confusion and have adjusted the figure in an attempt to clarify that the difference is exposure to BSA or SVEP1.

6. In the SVEP1 KO studies from Fig 5 and Fig S4, the authors call this a 'diet-induced diabetes' model, but seem to have only fed western diet in their studies. Were these mice on a different background (e.g. db/db, etc) or was this not a classic diabetes model? Furthermore, the authors state there was no metabolic impact of loss of this gene. However, the introduction also states that genetic variation in SVEP1 is associated with diabetes. Do the authors have an idea of why the mouse model doesn't reflect the human genetics?

Author reply: Our model employed prolonged feeding of high-fat Western diet to C57BL/6J mice which causes both diet-induced obesity and diabetes². We³ and others⁴ have shown SVEP1 is causally related to type 2 diabetes in humans, and we were surprised that SVEP1 depletion did not result in altered risk of diabetes in our mouse model. It is very challenging to speculate about why deletion of SVEP1 in mice does not result in the cardiometabolic phenotypes expected based on human data. Potential explanations include the inadequacy of mice to perfectly model human disease⁵, differences in the time-course of chronic disease development between humans and mice, the potential importance of SVEP1 prior to adulthood, outbred

² Surwit RS, Kuhn CM, Cochrane C, et al. Diet-induced type II diabetes in C57BL/6J mice. *Diabetes*. 1988 Sep;37(9):1163-7.

³ Jung IH, Elenbaas JS, Alisio A, et al. SVEP1 is a human coronary artery disease locus that promotes atherosclerosis. *Sci Transl Med*. 2021 Mar 24;13(586):eabe0357.

⁴ Emilsson V, Gudmundsdottir V, Gudjonsson A, et al. Coding and regulatory variants are associated with serum protein levels and disease. *Nat Commun*. 2022 Jan 25;13(1):481.

⁵ Ioannidis JP. Extrapolating from animals to humans. *Sci Transl Med*. 2012 Sep 12;4(151):151ps15.

(human) versus in-bred (mouse) populations that are genetically distinct, among others. Prior evidence demonstrates that *PEAR1* may play a less prominent role in mice than in humans⁶, raising the possibility that *SVEP1* may behave in a similar fashion. We cannot exclude any of these possibilities and believe to have qualified the claims in the manuscript to acknowledge this uncertainty.

7. Along the same lines, the authors state that BP is linked to *SVEP1* in humans, but no difference was reported in Fig S5 – can the authors please comment?

Author reply: Please see response to point 6 above.

8. In Fig 6, why is CVD used as the phenotype, rather than one more specifically related to thrombosis/platelet activation. Especially because the link between *SVEP1* and CAD has previously been described (Ref 4), it would be good to show some link to a clinically relevant clotting disorder (e.g. DVT, PE, etc). I think this could be important given the risk of pleiotropy with a phenotype as broad as CVD, especially for a factor that is linked to a number of cardiometabolic phenotypes in humans.

Author reply: We agree with the Reviewer that a) deep vein thrombosis (DVT) and pulmonary embolism (PE) have partially overlapping genetic signals with platelet activation *in vivo*; and b) the link between *SVEP1* and CAD has been previously demonstrated. However, in Figure 6 we chose CAD as the outcome for the Mendelian Randomization analysis specifically *because* of the prior link between *SVEP1* and CAD in order to test the hypothesis that *PEAR1* is causally related to CAD in a manner concordant with *SVEP1*. We are not aware of human genetic evidence that supports a strong association between either *PEAR1* or *SVEP1* with venous thromboembolism (VTE) despite robust associations of both genes with platelet reactivity *in vitro*, perhaps reflecting a portion of the difference between the genetic architectures of arterial and venous disease. A recently published excellent large-scale meta-analysis focused on the genetic basis of VTE⁷ did not identify association with VTE in either *PEAR1* or *SVEP1* despite leveraging data from >80,000 cases. In fact, approximately only half of shared loci between VTE and platelet parameters were concordant, reaffirming partially overlapping yet distinct genetic architectures. We hope that future studies will build upon our work and provide clarity on the mechanisms driving the genetic associations (or lack thereof) between *SVEP1* and *PEAR1* with human diseases.

Minor:

- In the introduction, the text describes an association for a variant with CAD risk as being shown in Fig 1A. However, that figure does not describe such an association and instead shows the structure of the gene. This reviewer suggests rephrasing of these sentences.

Author reply: We apologize for the prior confusion. The introduction text highlights *SVEP1* variants (p.D2702G and p.R229G) that are depicted above the cartoon *SVEP1* schematic in Figure 1A. We have attempted to clarify this in the figure legend.

- Are the blue circle and square in the figure legend for Fig 5J extraneous?
- The color scheme of Figs 5K and 5L are somewhat difficult for this reviewer to resolve and might be enhanced with different colors for clarity.

⁶ Criel M, Izzi B, Vandenbrielle C, Liesenborghs L, et al. Absence of *Pear1* does not affect murine platelet function *in vivo*. *Thromb Res*. 2016 Oct;146:76-83.

⁷ Thibord F, Klarin D, Brody JA, et al. Cross-Ancestry Investigation of Venous Thromboembolism Genomic Predictors. *Circulation*. 2022 Oct 18;146(16):1225-1242.

Author reply: We thank the Reviewer for pointing out the challenges with visually resolving these images. We have made several improvements to Fig 5J-L. The circle and square in Fig. 5J were added intentionally to maintain consistency with panels K-L and emphasize that the data represent the aggregation of platelets after the addition of SVEP1. We hope this is now clear and are happy to make additional adjustments if further clarity is needed.

Reviewer #4

I have no real comment to make on all the expression studies, interaction studies, platelet signalling studies, PEAR1 and Akt phosphorylation studies in the various cells, not do I find shortcomings in the adhesion studies and interaction studies between PEAR1 and SVEP1. Except for the few minor comments made below, I agree with every statement made in this paper. I want to congratulate the authors for such a comprehensive study, which I rarely met during a first round of submission.

Author reply: We thank Reviewer 4 for their comprehensive review of our manuscript and for these comments.

I just have 3 small, minor things:

1. The authors mention in the discussion that FcEpsilonR1alpha is not a physiological ligand because it does not activate platelets. That is only correct for the monomeric recombinant FcEpsilonR1alpha chain, but its pentameric equivalent (as it is present on immune cells) does trigger potent platelet activation via PEAR1. Whereas it is most probably correct that FcEpsilonR1alpha is not the prime ligand for PEAR1, the interaction is not nonproductive and can lead to human platelet activation, be it much less to murine platelet activation.

Author reply: We thank the Reviewer for highlighting this point and have edited the Discussion points around FcεR1α to incorporate the above points.

2. This study would benefit from some immunohistochemical staining of SVEP1 in basement membrane or in cross-sections of arterial tissue, to illustrate its localisation in subendothelial space vs media and/or adventitia (this may be somewhat redundant with pictures published before, but the emphasis should be put on the compartmental analysis, as part of the activation mechanism of endothelial cells and/or platelets, during injury, atherosclerosis).

Author reply: We agree with the Reviewer that immunohistochemical staining of SVEP1 in situ would provide great insight into the function of SVEP1. This has been an active area of interest of ours for several years; however, we still lack the reagents necessary to perform these experiments. The modular structure of SVEP1 and its high domain conservation with other extracellular matrix proteins have made developing a specific antibody very challenging. Although there have been reports of SVEP1 staining in the literature, we have been unable to replicate the staining when using appropriate controls (such as knockout models).

3. Whereas the strength and specificity of Mendelian Randomisation is well discussed (Discussion), this has not been done for the INTERVAL approach. It would be informative in the Supplement or the Methods to read a few lines on the specificity and strength of this methodology. Such would facilitate interpreting strength and analysis in Figure 1.

Author reply: We believe the Reviewer is referring to the specificity and strength of plasma protein levels as measured by the SOMAScan platform. Although we will defer to the Editor's opinion, we feel this is outside of the scope of the current manuscript as it has been extensively discussed in the manuscripts describing the generation and analysis of SOMAScan data in cohort studies. Many of these studies are cited within this manuscript.

REVIEWER COMMENTS

Reviewer #1 (Remarks to the Author):

I thank the investigative team for thoroughly thinking about and responding to the series of comments I included in my initial review. I have enjoyed reading the first draft of this manuscript as well as the current iteration.

I do believe the manuscript would benefit from the authors including the data that they generated as a result of my first two comments (i.e. the conditional analysis figure [which could be included in Supplementary Figure 1] as well as results obtained using the GREML approach for the SVEP1 and PEAR1 instruments. However, I will defer to the editorial team whether or not they believe that these would be meaningful inclusions.

I have no more comments. Best wishes.

Reviewer #2 (Remarks to the Author):

Thank you for the addition of negative controls for the BLI experiment. With these, I believe the BLI experimentations showing an interaction between SVEP1 and PEAR1 are valid and worthy of publication.

Reviewer #3 (Remarks to the Author):

Thank you for the thorough response to the queries from myself and the other reviewers. The manuscript is improved, but I remain unclear if the rs147639000 variant alters SVEP1 binding/stabilization/degradation/etc, or the molecular mechanism by which such as trans-pQTL effect could be occurring. Regarding the human MR studies, I appreciate the point that platelet parameters and VTE don't always correlate in human genetic studies. But was there any trend in the studies you mention in the rebuttal? If not, what mechanism do the authors invoke for the phenotypes which were observed in humans (Fig 6A/B for platelet count and volume), which were not seen in the mice (Fig 5A)? Do alterations in platelet reactivity (as shown in mice) lead to these clinical phenotypes? While the CVD link is interesting and hasn't been described for PEAR1, it has been described for SVEP1 which appears to have a significant number of pleiotropic cardiometabolic effects, making it difficult to suggest this pathway alters platelet biology in humans. Perhaps platelet reactivity assays in primary cells from carriers of risk allele would help assuage those concerns.

Reviewer #4 (Remarks to the Author):

The authors have incorporated new evidence and have clarified a number of items, which convincingly have increased the understanding of the claims made. Furthermore, the description of activation studies for the PEAR1-SVEP1 interaction has made a very credible and convincing case. I congratulate the authors for an interesting study.

Reviewer #5 (Remarks to the Author):

The authors performed a very thorough and systematic evaluation to characterize the interactions between SVEP1 with PEAR1 using several approaches. This review is specifically focused on proteomics workflow in the study. Pull down (proximity- and affinity-based) assays were performed followed by mass spectral analysis using both timsTOF and Orbitrap MS instruments. Both the instruments are high throughput for protein identification and characterization studies. The findings revealed that SVEP1 may integrate with basement membrane and suggested possible interactions with several PEAR1-expressing cells. While the findings are interesting, there are few comments about the approach that needs to be addressed.

1. Are the 8 proteins listed in figure 2G identified in both approaches on timsTOF and Orbitrap MS? How many technical replicate experiments were performed, and did they identify in all of them? Two experiments were performed using each approach, but there is a high and low flow rate analysis using Orbitrap MS, which is considered as replicate experiment? Why low flow rate analyses were performed? Did you see any increased protein identifications/coverage or different protein identifications?

2. Why are SVEP1 and PEAR1 not identified in MS experiments? Did they not identify or they did not qualify within the $p < 0.1$ threshold? Proteins that big are expected to identify with at least ≥ 1 unique peptide(s).

3. Full images of immunoblots especially for Figure 2 and Figure S2 would be helpful for better evaluation.

4. Lines 744, 745: Why only 30°C is used for LysC and trypsin digestion while 37°C is the optimum temperature for digestion protocols? And again, 37°C is used later on proteins from antibody beads (lines 779 and 780).

5. TFA is usually avoided when the sample is analyzed on LCMS instruments as they can lead to signal suppression and also it is very tedious (sometimes almost impossible) to get rid of TFA generated ions. Though evaporated, it is usually not preferable to avoid any potential problems. Formic acid can be used instead with almost equal success in eluting peptides and does not interfere with MS signals.

Minor comments:

6. Line 847: MS raw data is from Orbitrap MS? Mention the model of the instrument as well.

7. Figure 2 legend: Line 1189: list of proteins enriched in experiments represented in 'F' not 'E'.

8. Line 894: Use mass 'spectrometry' not 'spectroscopy'.

9. Use full term for acronyms TFA, FA, TIMS, PASEF etc. These might be common terms in proteomics community, but it is good to provide details for those new to this field.

Response to Reviewer's Comments

We thank the first four Reviewers for evaluation of our revised manuscript and the fifth Reviewer for their review of our mass spectrometry-based experiment. In response to the Reviewer's comments we have revised the text to clarify additional points as detailed below.

Reviewer #1

I thank the investigative team for thoroughly thinking about and responding to the series of comments I included in my initial review. I have enjoyed reading the first draft of this manuscript as well as the current iteration.

I do believe the manuscript would benefit from the authors including the data that they generated as a result of my first two comments (i.e. the conditional analysis figure [which could be included in Supplementary Figure 1] as well as results obtained using the GREML approach for the SVEP1 and PEAR1 instruments. However, I will defer to the editorial team whether or not they believe that these would be meaningful inclusions.

I have no more comments. Best wishes.

Author reply: We appreciate the Reviewer's evaluation of our revised manuscript and thank them for their prior helpful comments. We agree with the Reviewer and have added these results to the manuscript as suggested in the text and in a revised Supplemental Figure 1.

Reviewer #2

Thank you for the addition of negative controls for the BLI experiment. With these, I believe the BLI experimentations showing an interaction between SVEP1 and PEAR1 are valid and worthy of publication.

Author reply: We thank the Reviewer for their review of our revised manuscript and for their prior helpful suggestions regarding our BLI experiment.

Reviewer #3

Thank you for the thorough response to the queries from myself and the other reviewers.

Author reply: We appreciate the Reviewer's evaluation of our revised manuscript and thank them for their prior helpful comments.

The manuscript is improved, but I remain unclear if the rs147639000 variant alters SVEP1 binding/stabilization/degradation/etc, or the molecular mechanism by which such as trans-pQTL effect could be occurring.

Author reply: We agree with the Reviewer that the mechanism by which the rs147639000 (PEAR1 p.D343N) variant alters SVEP1 levels is not completely clear. Our BLI results shown in Figures S2B and S2C demonstrate that PEAR1 D343N may have lower affinity to SVEP1 compared to WT PEAR1 (1.76nM versus 0.708nM, respectively) which is directionally consistent with the hypothesis that PEAR1 D343N results in increased plasma SVEP1 via decreased receptor affinity. However, we recognize that there are inherent differences between the in vitro assay and in vivo human physiology which preclude a definitive conclusion that the trans pQTL effect is a result of altered affinity. We hope to explore this possibility in future work.

Regarding the human MR studies, I appreciate the point that platelet parameters and VTE don't always correlate in human genetic studies. But was there any trend in the studies you mention in the rebuttal? If not, what mechanism do the authors invoke for the phenotypes which were observed in humans (Fig 6A/B for platelet count and volume), which were not seen in the mice (Fig 5A)? Do alterations in platelet reactivity (as shown in mice) lead to these clinical phenotypes? While the CVD link is interesting and hasn't been described for PEAR1, it has been described for SVEP1 which appears to have a significant number of pleiotropic cardiometabolic effects, making it difficult to suggest this pathway alters platelet biology in humans. Perhaps platelet reactivity assays in primary cells from carriers of risk allele would help assuage those concerns.

Author reply: We agree with the Reviewer that the relationship between the SVEP1 / PEAR1 signaling axis and human disease remains a fruitful area of study. Unfortunately, the summary statistics for the study mentioned in our prior rebuttal are not yet publicly available so we are unable to determine if there was a trend for association with genetic variation in *SVEP1* or *PEAR1* with venous thromboembolism. Although incompletely characterized, it appears that murine PEAR1 may play a less prominent role in platelet function than human PEAR1¹ which may in part account for the differing phenotypes observed in humans and mice. We hope future studies of SVEP1 and PEAR1 will provide additional insights into their roles in platelet biology. Finally, we agree that platelet reactivity assays in primary cells from carriers of the risk allele would be interesting but recruiting these individuals would represent a substantial undertaking (PEAR1 D343N is carried on ~2% of European chromosomes and the maximum population allele frequency is only ~7%); as a result, it falls outside of the scope of the current manuscript.

¹ Criel M, Izzi B, Vandenbrielle C, Liesenborghs L, et al. Absence of Pear1 does not affect murine platelet function in vivo. *Thromb Res.* 2016 Oct;146:76-83.

Reviewer #4

The authors have incorporated new evidence and have clarified a number of items, which convincingly have increased the understanding of the claims made. Furthermore, the description of activation studies for the PEAR1-SVEP1 interaction has made a very credible and convincing case. I congratulate the authors for an interesting study.

Author reply: We appreciate the Reviewer's evaluation of our revised manuscript and thank them for their prior helpful comments.

Reviewer #5

The authors performed a very thorough and systematic evaluation to characterize the interactions between SVEP1 with PEAR1 using several approaches. This review is specifically focused on proteomics workflow in the study. Pull down (proximity- and affinity-based) assays were performed followed by mass spectral analysis using both timsTOF and Orbitrap MS instruments. Both the instruments are high throughput for protein identification and characterization studies. The findings revealed that SVEP1 may integrate with basement membrane and suggested possible interactions with several PEAR1-expressing cells. While the findings are interesting, there are few comments about the approach that needs to be addressed.

Author reply: We thank Reviewer 5 for their review of our experiment using mass spectrometry to identify secreted proteins that might interact with SVEP1.

1. Are the 8 proteins listed in figure 2G identified in both approaches on timsTOF and Orbitrap MS? How many technical replicate experiments were performed, and did they identify in all of them? Two experiments were performed using each approach, but there is a high and low flow rate analysis using Orbitrap MS, which is considered as replicate experiment? Why low flow rate analyses were performed? Did you see any increased protein identifications/coverage or different protein identifications?

Author reply: We thank the Reviewer for these comments. Two independent experiments were performed under each approach utilizing the services of a University Mass Spectrometry core facility; all eight proteins listed in Figure 2G were identified in all independent experiments. The primary specification that the core facility uses to qualify instrument performance is the number of identified peptides from a standard HeLa digest (~20,000+/- 2000). This test is performed by the core facility before sample analysis and after every 8 samples (~16hours). The flow rates are varied slightly to accommodate the higher pressure from column ageing without affecting the HeLa specification. The replicate samples in this study were acquired on different days and stored at -80C between acquisitions which may also contribute to any differences. However, the core facility has found (as have other labs) that the method of data acquisition has a larger impact; these discovery data were acquired using data-dependent LC-MS. This mode of acquisition selects the 8 most intense precursor ions from MS1 for tandem MS. It is well-recognized that this stochastic nature of data acquisition results in discordance of results even when the samples are injected repetitively on the same day.

2. Why are SVEP1 and PEAR1 not identified in MS experiments? Did they not identify or they did not qualify within the $p < 0.1p$ threshold? Proteins that big are expected to identify with at least ≥ 1 unique peptide(s).

Author reply: We thank the Reviewer for these comments. This experiment was designed to test interaction of SVEP1 with secreted proteins. As a cell-surface receptor, PEAR1 is not present in the enriched media in sufficient quantities to identify its binding to SVEP1 by MS. As the bait protein, SVEP1 was identified in both approaches and was enriched in each experimental sample relative to its control sample. We did not report this result because it is the dependent variable in the experiment.

3. Full images of immunoblots especially for Figure 2 and Figure S2 would be helpful for better evaluation.

Author reply: We apologize if the Reviewer was not provided with the file containing full immunoblots from all Figures which was included in our original submission. We include full immunoblots for Figure 2 and Figure S2 below for the Reviewer's convenience.

Full unedited blot for Figure 2B
2021-09-29

+ - Lane included in figure

PEAR1
*Note different exposures
for input and pulldown

SVEP1
(Myc)

Full unedited blot for Figure 2C
2021-09-23

+ - Lane included in figure

PEAR1

SVEP1
(Myc)

Full unedited blot for Figure 2E
2021-12-21

PEAR1

beta-Tubulin

Full unedited blot for Figure S2D
2021-12-16

+ - Lane included in figure

SVEP1
(Myc)
*Note different exposures for
input and pulldown

Streptavidin

Full unedited blot for Figure S2F
1-21-2021

+ - Lane included in figure

Fibronectin
(low exposure)

Fibronectin
(high exposure)

4. Lines 744, 745: Why only 30oC is used for LysC and trypsin digestion while 37oC is the optimum temperature for digestion protocols? And again, 37oC is used later on proteins from antibody beads (lines 779 and 780).

Author reply: The core facility relayed that they utilized the lower temperature in order to reduce the extent of peptide carbamylation in urea endoprotease digests². For the other protocol that used 37C, the urea was removed using the filter aided sample preparation (FASP) method.

5. TFA is usually avoided when the sample is analyzed on LCMS instruments as they can lead to signal suppression and also it is very tedious (sometimes almost impossible) to get rid of TFA generated ions. Though evaporated, it is usually not preferable to avoid any potential problems. Formic acid can be used instead with almost equal success in eluting peptides and does not interfere with MS signals.

Author reply: Although we appreciate the Reviewer's comment surrounding different options for preparing peptides prior to mass spectrometry, these analyses were performed in a core facility and we were unable to specify the methods used to prepare the peptides. We do not believe the use of TFA would be expected to generate false positives in mass spectrometry and do not believe the conclusions we have drawn from these data are affected by its use.

Minor comments:

6. Line 847: MS raw data is from Orbitrap MS? Mention the model of the instrument as well.

Author reply: We thank the Reviewer for noticing this. The data were acquired using a hybrid-quadrupole-Orbitrap mass spectrometer (Q-Exactive Plus, Thermo Fisher) which we now specify in the text.

7. Figure 2 legend: Line 1189: list of proteins enriched in experiments represented in 'F' not 'E'.

Author reply: We thank the Reviewer for identifying this typographical error which has been corrected.

8. Line 894: Use mass 'spectrometry' not 'spectroscopy'.

Author reply: We thank the Reviewer for identifying this typographical error which has been corrected.

9. Use full term for acronyms TFA, FA, TIMS, PASEF etc. These might be common terms in proteomics community, but it is good to provide details for those new to this field.

Author reply: We thank the Reviewer for this suggestion and have provided full terms for these acronyms upon their first use in the manuscript.

² Mertins et al., Reproducible workflow for multiplexed deep-scale proteome and phosphoproteome analysis of tumor tissues by liquid chromatography–mass spectrometry. Nat Protoc. 2018 Jul;13(7):1632-1661.